# A systematic comprehensive longitudinal evaluation of dietary factors associated with acute myocardial infarction and fatal coronary heart disease

Soodabeh Milanlouei [1,5], Giulia Menichetti [1,5], Yanping Li [2], Joseph Loscalzo [3], Walter C. Willett[2,3] & Albert-László Barabási [1,3,4 ✉]

Environmental factors, and in particular diet, are known to play a key role in the development of Coronary Heart Disease. Many of these factors were unveiled by detailed nutritional epidemiology studies, focusing on the role of a single nutrient or food at a time. Here, we apply an Environment-Wide Association Study approach to Nurses' Health Study data to explore comprehensively and agnostically the association of 257 nutrients and 117 foods with coronary heart disease risk (acute myocardial infarction and fatal coronary heart disease). After accounting for multiple testing, we identify 16 food items and 37 nutrients that show statistically significant association – while adjusting for potential confounding and control variables such as physical activity, smoking, calorie intake, and medication use – among which 38 associations were validated in Nurses' Health Study II. Our implementation of Environment-Wide Association Study successfully reproduces prior knowledge of diet-coronary heart disease associations in the epidemiological literature, and helps us detect new associations that were only marginally studied, opening potential avenues for further extensive experimental validation. We also show that Environment-Wide Association Study allows us to identify a bipartite food-nutrient network, highlighting which foods drive the associations of specific nutrients with coronary heart disease risk.

[1] Center for Complex Network Research, Northeastern University, Boston, MA, USA. [2] Department of Nutrition, Harvard T.H. Chan School of Public Health, Boston, MA, USA. [3] Channing Division of Network Medicine, Department of Medicine, Brigham and Women's Hospital, Boston, MA, USA. [4] Center for Network Science, Central European University, Budapest, Hungary. [5]These authors contributed equally: Soodabeh Milanlouei, Giulia Menichetti. ✉email: alb@neu.edu

The prevalence of heart disease, the leading cause of death throughout the world, is strongly influenced by diet and eating habits[1–4]. For example, a recent CDC (Centers for Disease Control and Prevention) report[5], focusing on death rates caused by heart disease across the United States, documented substantial regional differences compatible with different eating patterns. Similarly, while among individuals of Japanese descent, coronary heart disease (CHD) incidence rates are only 1.6 per person-years in Japan, it increases to 3.0 in Hawaii, and 3.7 in San Francisco[6], differences that cannot be explained by genetic factors, documenting the key role dietary and other environmental factors play in the development of the disease.

Much of our knowledge about the role of food on health comes from epidemiological association studies in which a single or limited number of exposure(s) is/are analyzed in relation to a phenotype, representing a hypothesis-driven path towards understanding diet–disease relationships. Yet, diet is not a simple sum of several nutrients, as each food product consists of a mixture of nutrients associated with multiple compounds of limited or unknown nutritional value[7]. Accordingly, the effect of each dietary compound on human health should not be investigated in isolation, but in the presence of other associated chemical compounds and relevant food sources. For example, Kolonel et al.[8] initially reported that beta-carotene consumption was positively associated with the risk of prostate cancer. While this finding was worrisome, in a subsequent analysis of foods, intake of carrots, the largest source of beta-carotene, was not related to the risk of prostate cancer; the observed association was due to intake of papaya[9]. Thus, the analysis of foods provided evidence against the effect of beta-carotene and suggested that some factors specific to papaya might be responsible for the original finding[10]. An alternative approach is dietary pattern analysis that focuses on the effects of the overall diet[11–13], rather than a single or a few nutrients. While dietary pattern analyses are ideal in the development of nutritional guidelines[14], they are insufficient for the agnostic discovery of new signals for further experimental or mechanistic validation.

As an alternative to the traditional epidemiological studies, environment-wide association studies (or EWAS) were proposed to identify new environmental factors in disease and disease-related phenotypes in an unbiased manner. EWAS is inspired by the analytical procedures developed in genome-wide association studies (GWAS)[15] in which a panel of "exposures" (genotype variants) is studied in relation to a phenotype of interest. For example, using the National Health and Nutrition Examination Survey dataset, an EWAS study explored the associations of 543 environmental attributes with type 2 diabetes, identifying five statistically significant associations validated across independent cohorts[15]. Wulaningsih et al.[16] investigated 182 nutrition and lifestyle factors in relation to abdominal obesity, finding a statistically significant association of obesity with five factors in men and seven factors in women. Merritt et al.[17] used European Prospective Investigation into Cancer and Nutrition (EPIC) data to evaluate endometrial cancer risk associations for the dietary intake of 84 foods and nutrients, concluding that only coffee intake had a statistically significant inverse relationship.

Despite the recent success of the EWAS methodology in unveiling multiple nutritional factors that together may contribute to our health, its widespread use is undermined by several factors and limitations[18]. Indeed, failing to achieve adequate statistical power in association detection, EWAS studies could not always recover known environment–disease associations confirmed by large, prospective cohort studies and randomized trials[17,19,20]. As we show below, these failures are not inherent in the EWAS methodology, but are mainly rooted in the limited size, limited variability, and lack of repeated measurements of the

datasets to which EWAS has been applied thus far. While the statistical power of the EWAS study approach is a legitimate concern, the magnitude of the statistical power depends on multiple factors, including the nature and the size of the dataset, as well as the statistical tools/models used for the analysis. Indeed, as we show here, if we apply a wide-association study approach to an adequately sized longitudinal cohort dataset with sufficient variability, we consistently recover prior knowledge about diet–disease relationships.

The EWAS methodology may be particularly useful for diseases for which nutritional associations are unknown[18]. While the effect of dietary exposures on heart disease has been extensively studied and the causal effects of many of these associations confirmed, the diet–disease literature occasionally demonstrates conflicting findings[2,21–23], limiting our understanding of the true effect of dietary exposures on diseases. We will show that the wide-association study approach can provide comparable insights in an efficient manner by applying an unbiased standardized set of analytical tools.

Here we implement an EWAS methodology, aiming to identify dietary factors associated with CHD systematically and comprehensively, focusing on both nutrient intake and food consumption. To overcome the limited statistical power of previous studies, we apply our methodology to the Nurses' Health Study (NHS), a longitudinal prospective study designed to investigate the longitudinal effects of nutrition on health and disease development. While there are larger cohorts available for studying chronic diseases, such as the UK Biobank[24] and the China Kadoorie Biobank[25], both with around 500,000 participants, NHS is unique, owing to comprehensive longitudinal dietary data collection. Beginning in 1976, NHS gathered registered female nurses, ages 30–55 years, from across the United States, initially designed to investigate the use of oral contraception in relation with risk of breast cancer. Participants are asked to complete questionnaires every 2 years, and in 1980 a Food Frequency Questionnaire (FFQ) has been included, designed to capture dietary behaviors. Follow-up dietary questionnaires were administered in 1984, 1986, and every 4 years since then. Questionnaires used from 1984 and thereafter included about 130 foods plus detailed information about brands and types of margarine, breakfast cereals, multiple vitamins, and types of fat used for cooking and baking. As health professionals, nurses were chosen for their ability to complete the health-related questionnaires thoroughly and accurately[26]. To date, NHS has been expanded to NHS II and NHS III to cover a younger population of nurses[27]. These three cohorts resulted in an extensive published body of research on the relationships of environmental and genetic factors to various diseases[28]. The dietary drivers of CHD have been extensively studied within NHS data, most analyses primarily focusing on a single or limited number of exposures, while controlling for an appropriate set of adjusting variables. Some of these findings, as those for trans-fats, have inspired experimental studies and were confirmed to have a causal effect on developing heart diseases[29–31].

To have a broad picture of the existing knowledge about diet–disease associations in the NHS data, we mined the literature to identify all studies exploring the dietary determinants of heart-related diseases in original NHS and successive cohorts. The resulting knowledge graph (Fig. 1) shows that the most extensively studied cardiovascular phenotype using NHS data is CHD. Here, we use the term *negative* when a higher level of exposure is associated with a lower CHD risk. Similarly, we use *positive* term when a higher level of an exposure is associated with CHD risk. We made this choice for simplicity, and it should not be confused with a causal relationship.

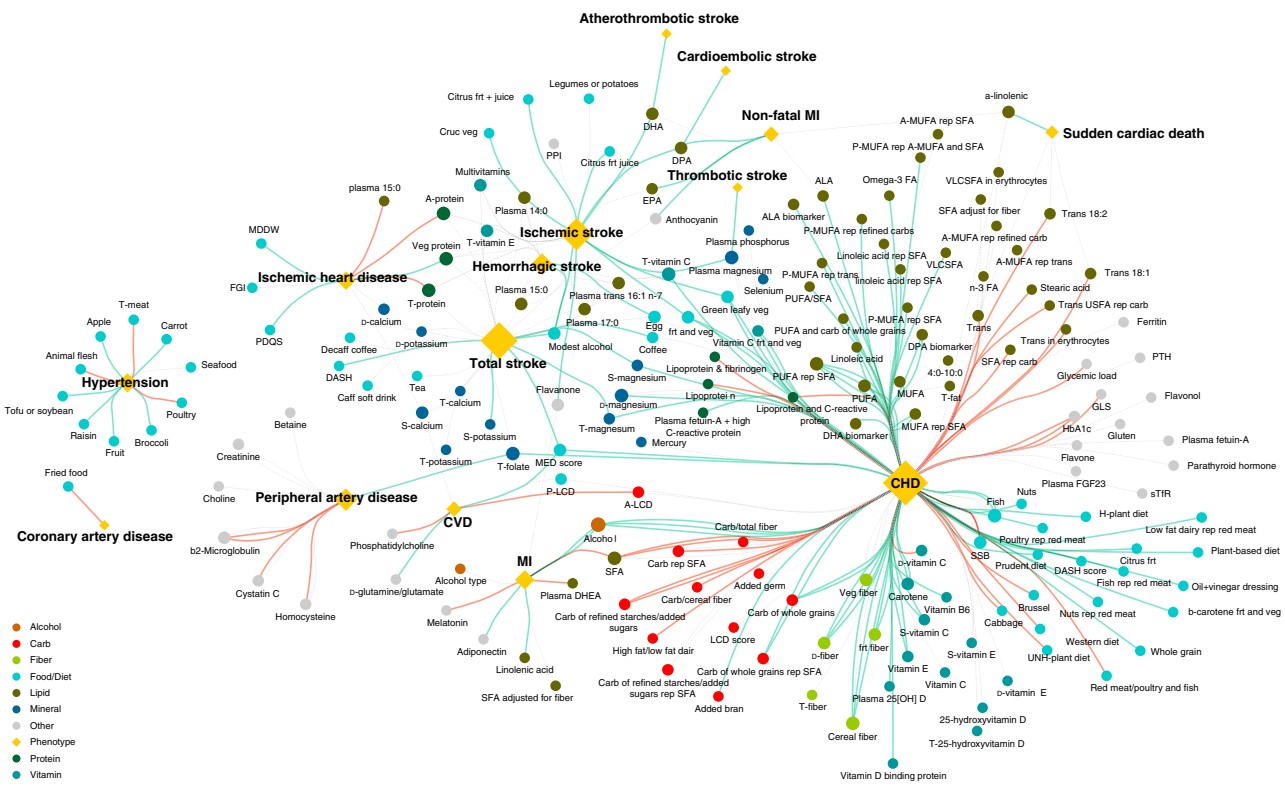

**Fig. 1 The Knowledge Graph of the Dietary Factors Associated with Cardiovascular Disease in the NHS Data.** The nodes of the graph are dietary exposures (circles) and cardiovascular diseases (diamonds) as two sets of nodes. Each studied association is shown by an edge whose color denotes the "direction" (i.e., positive or negative) of the association. Green and red links indicate negative and positive associations, respectively, between an exposure and a disease. Grey links denote associations that were studied but not found to be statistically significant. In the context of NHS, CHD refers to non-fatal MI and fatal coronary heart disease; also, coronary artery disease (CAD) refers to non-fatal MI and fatal coronary artery disease. Cardiovascular disease (CVD) is defined as a composite of coronary artery disease and non-fatal or fatal stroke. Source data are provided in Source Data - Figure 1.xlsx. The figure aims to illustrate the body of work derived from NHS data on cardiovascular diseases. A detailed comparison of the result of our analysis with previous NHS-related work and other findings in the literature is provided in the section "Comparison with the literature".

Excluding studies of biomarkers and tissues, the neighborhood of CHD comprises nutrients, food items, and dietary scores. Cumulatively, 120 associations were studied, documenting 63 negative associations and 22 positive relations with CHD; in the remaining cases, there was no significant association between dietary exposure and CHD risk, as in the case of dietary magnesium[32]. The space of studied exposures is rather heterogeneous and is often driven by either the researcher's interests or evidence from animal or mechanistic studies.

As illustrated in the knowledge graph, the single-association studies using NHS data have broadened our understanding of the dietary determinants of CHD. Some non-significant associations were found to be significant after the application of new statistical approaches in larger datasets. For example, while a study using the original NHS data found no association between fruit fiber intake and risk of CHD[33], a pooled analysis of three NHS cohorts found that the higher consumption of dietary fiber from fruits was negatively associated with risk of CHD[34].

Here, we show that a wide-association approach allows us to scan efficiently and systematically the dietary determinants of CHD, bypassing the problem with missing significant associations in epidemiological studies. By applying EWAS methodology to the NHS data, we find that a wide-association approach not only recovers the existing knowledge on diet–disease association, but also facilitates the discovery of novel associations, potentially inspiring future follow-up studies.

## Results

**Main findings**. During the follow-up period, 2774 incident cases of non-fatal MI or fatal CHD were documented in NHS. The baseline average of total caloric intake and body mass index (BMI) among participants who later developed CHD were slightly higher than in those who did not develop CHD. In addition, prospective case subjects on average had lower physical activity compared with the non-case population (Supplementary Table 1). We examined the effect of 374 exposures on CHD risk, including 257 nutrients and 117 food items. The descriptive characteristics of these exposures are shown in Supplementary Table 2.

For the first phase of EWAS, using Cox regression models we collected the estimated effect size, the variance of effect size, hazard ratio (HR) for one standard deviation, $P$ value, the 95% confidence interval (CI) for the HR, $P$ value regarding the proportionality assumption, and the variance inflation factor (VIF). Consider, for example, the results of the long-term effect of isorhamnetin—an *O-methylated flavon-ol* from the class of flavonoids (Fig. 2b). The estimated HR (0.91, 95% CI: 0.87–0.95; $P$ value $1.59 \times 10^{-5}$) implies that one standard deviation higher consumption of Box–Cox-transformed isorhamnetin is associated with 91% lower CHD risk. The $P$ value regarding the proportionality assumption indicates that the use of the Cox model is appropriate. The VIF equal to 1.27 suggests that there is no severe multicollinearity among the variables involved in the isorhamnetin test.

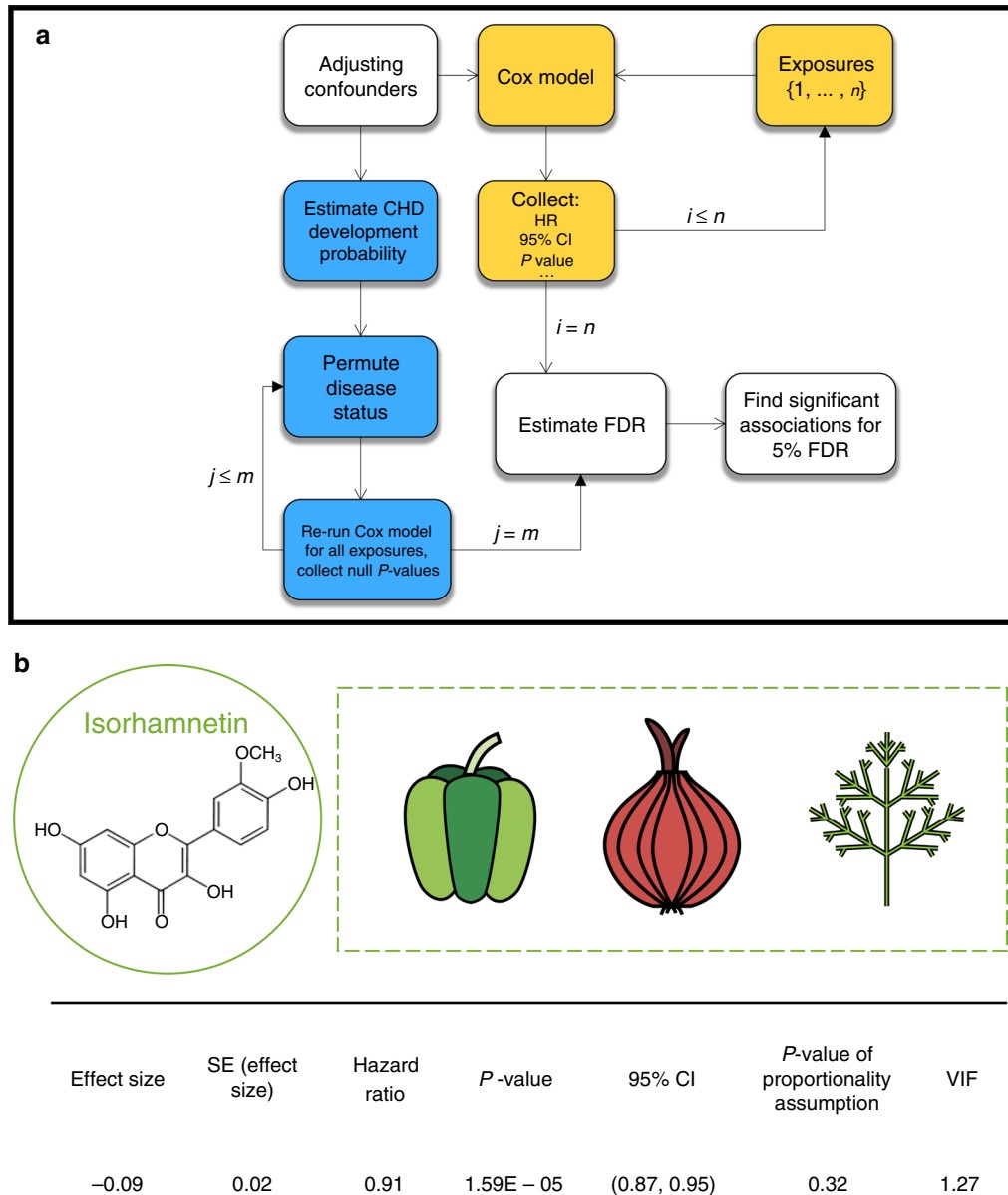

| Effect size | SE (effect size) | Hazard ratio | P -value | 95% CI | P-value of proportionality assumption | VIF |
|---|---|---|---|---|---|---|
| −0.09 | 0.02 | 0.91 | 1.59E − 05 | (0.87, 0.95) | 0.32 | 1.27 |

**Fig. 2 Ewas Methodology Description and Output. a** For each exposure (*i*), we fit a Cox model to estimate CHD risk, while controlling for a set of adjusting variables (*n* exposures, in total). Through this process, we also collect the VIF and the *P* value regarding the proportionality assumption. After the fitting phase (yellow), we proceed with the multiple testing protocols (blue). We leverage confounding variables to estimate the likelihood of CHD development for each subject, and the disease status is then permuted accordingly, *m* times. For each exposure, the FDR is estimated, and associations with an FDR < 0.05 are considered statistically significant. **b** Isorhamnetin is an *O-methylated flavonol* from the class of flavonoids mainly found in green pepper, red onion, and dill. EWAS shows that isorhamnetin intake is negatively associated with CHD risk (HR: 0.91; 95% CI: 0.87–0.95; *P* value 1.59 × 10$^{-5}$, from two-sided Wald test, with no adjustment for multiple comparisons). The *P* value regarding the proportionality assumption indicates the appropriateness of using the Cox model. The VIF of 1.27 is an indication of the absence of severe multicollinearity.

Figure 3a shows the distribution of Cox model *P* values for all investigated exposures. Exposures are ordered by the estimated HR, so that exposures with HR > 1.0 have harmful effects on CHD risk and exposures with HR < 1.0 are expected to be beneficial. Insignificant associations are mainly distributed around an HR of one, and exposures with smaller *P* values are scattered at the two ends of the distribution. Using the permutation procedure to account for multiple testing, we found 53 significant associations, including 16 food items and 37 nutrients. All significant associations had VIF < 5. For all significant associations, except for phytate, the *P* value regarding the proportionality assumption is >0.05. The minimum statistical power for detecting the smallest absolute effect size was 0.59, which is considered to be a moderate to high level of power in clinical studies. A list of exposures that have a statistically significant association with CHD risk, together with their estimated HRs, is shown in Table 1. A list of both significant and non-significant associations is provided in Supplementary Table 3. We analyzed the correlations among the significant exposures, the result of which is shown in Supplementary Fig. 3. To help interpret our findings, we calculated the HR of each quintile of exposure intake compared with the first quintile, as a reference group (Supplementary Table 6).

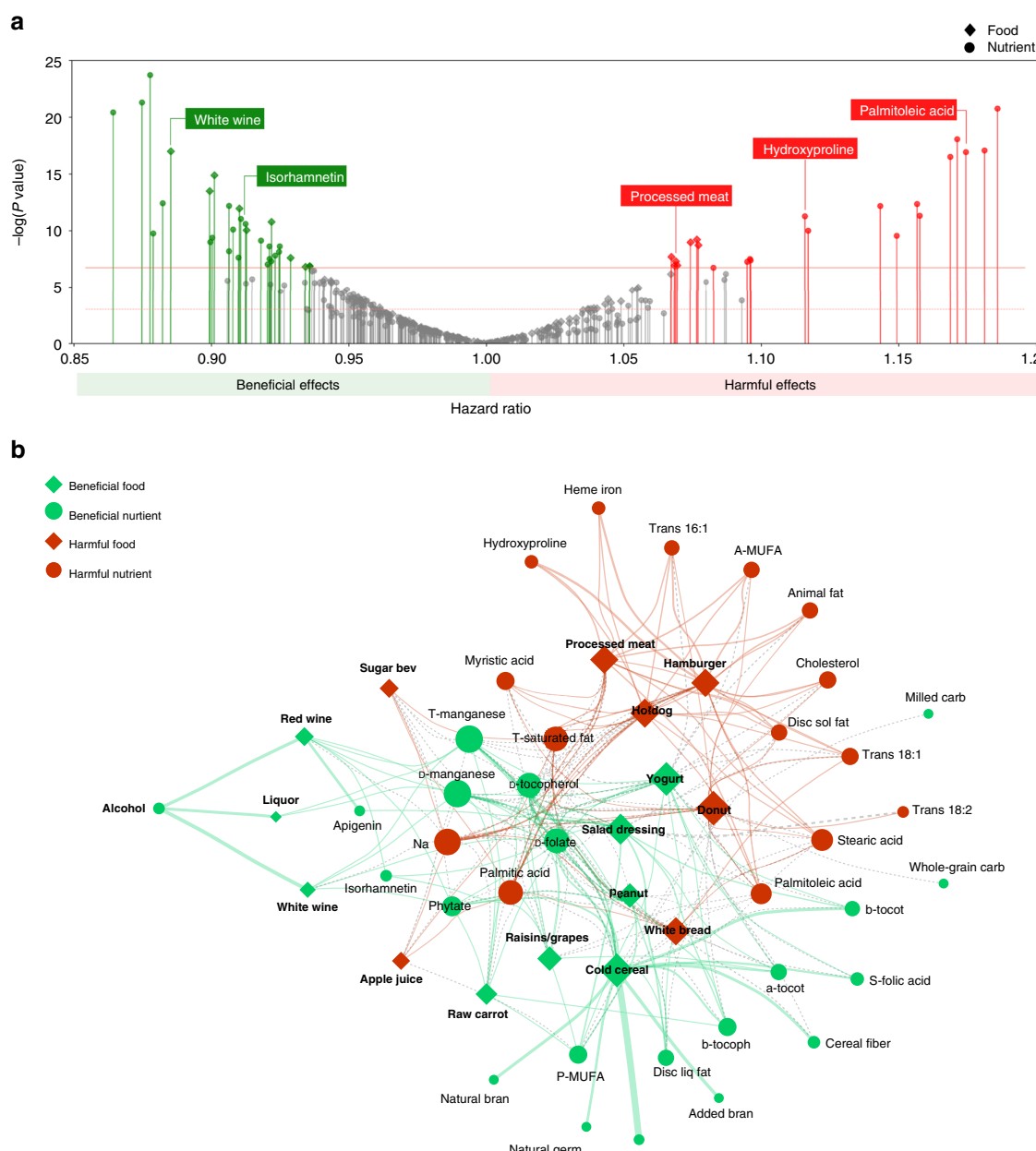

**Fig. 3 P value Distribution of Dietary Factors and the Bi-partite Food-Nutrient Network. a** *P* values are associated with two-sided Wald tests. The *Y* axis indicates the negative logarithm of the *P* value of each exposure. The dotted red horizontal line marks the level of 0.05 *P* value threshold and the solid red line demonstrates the level of 0.00117 *P* value threshold associated with the level of 0.05 FDR. Significant foods (diamonds) and nutrients (circles) with a negative association (HR < 1) are highlighted in green and exposures with a positive association (HR > 1) are shown in red. Source data are provided in Source Data - Figure 3-a.xlsx. **b** Bi-partite food-nutrient network, where negatively associated exposures are shown in green and positively associated exposures are shown in red. Different shapes are used to differentiate between foods (diamonds) and nutrients (circles). The node size is proportional to the absolute value of the estimated effect size. The link thickness is proportional to the amount of each nutrient's composition in food. Source data are provided in Source Data - Figs. 3-b and 4-b.xlsx.

Traditional epidemiological studies are limited to the detection of a single exposure (food or nutrient) in relation to CHD; however, the developed EWAS methodology allows us to explore the space of food/nutrient associations related to the disease. That is, in addition to unveiling which nutrient shows significant association with CHD, EWAS also helps us understand which nutrient in connection with which food is responsible for the effect. To demonstrate this principle, we use the food composition table of NHS to extract the contribution of each significant food to the total amount of a significant nutrient in the food supply

(Fig. 3a). Using a force-directed layout algorithm, we represent this information as a bipartite network, allowing us to explore the significant inter-dependencies among nutrients and food items. In Fig. 3b, negatively associated nutrients and foods are color coded as green and positively associated nutrients and foods are shown in red. We retrieve two clearly distinct clusters, negatively associated nutrients and foods on one hand, and positively associated nutrients and foods on the other. We also find that several food items, such as white bread and yogurt are connected to both negatively and positively associated nutrients. As

**Table 1 EWAS results.**

| Type | Exposure | Effect size | SE (effect size) | Hazard ratio | 95% CI | P value of PH | VIF | P value | FDR |
|---|---|---|---|---|---|---|---|---|---|
| Nutrient | Alcohol | −0.13 | 0.02 | 0.88 | (0.84, 0.91) | 0.19 | 1.12 | 4.98E-11 | 0.000 |
| Nutrient | Added bran from wheat, rice, etc. | −0.13 | 0.02 | 0.87 | (0.84, 0.91) | 0.1 | 1.22 | 5.56E−10 | 0.000 |
| Nutrient | Trans 16:1 | 0.17 | 0.03 | 1.19 | (1.12, 1.25) | 0.41 | 2.15 | 9.59E-10 | 0.000 |
| Nutrient | Discretionary liquid fat | −0.15 | 0.02 | 0.86 | (0.82, 0.91) | 0.84 | 1.48 | 1.33E-09 | 0.000 |
| Nutrient | Animal MUFA | 0.16 | 0.03 | 1.17 | (1.11, 1.24) | 0.78 | 2.09 | 1.42E-08 | 0.001 |
| Nutrient | Discretionary solid fat | 0.17 | 0.03 | 1.18 | (1.11, 1.25) | 0.71 | 2.51 | 3.82E-08 | 0.001 |
| Food | White wine | −0.12 | 0.02 | 0.89 | (0.85, 0.92) | 0.66 | 1.07 | 4.14E-08 | 0.001 |
| Nutrient | Palmitoleic acid | 0.16 | 0.03 | 1.17 | (1.11, 1.24) | 0.64 | 2.34 | 4.42E-08 | 0.001 |
| Nutrient | Animal fat | 0.16 | 0.03 | 1.17 | (1.10, 1.24) | 0.9 | 2.26 | 6.73E-08 | 0.001 |
| Food | Salad/oil and vinegar dressing | −0.1 | 0.02 | 0.9 | (0.87, 0.94) | 0.24 | 1.09 | 3.41E-07 | 0.001 |
| Food | Yogurt | −0.11 | 0.02 | 0.9 | (0.86, 0.94) | 0.09 | 1.07 | 1.37E-06 | 0.003 |
| Nutrient | Phytate | −0.13 | 0.03 | 0.88 | (0.84, 0.93) | 0.02 | 1.96 | 4.01E-06 | 0.005 |
| Nutrient | Stearic acid | 0.15 | 0.03 | 1.16 | (1.09, 1.23) | 0.82 | 2.76 | 4.25E-06 | 0.005 |
| Nutrient | Carbohydrate from milled wholegrain | −0.1 | 0.02 | 0.91 | (0.87, 0.95) | 0.29 | 1.29 | 5.03E-06 | 0.005 |
| Nutrient | Sodium | 0.13 | 0.03 | 1.14 | (1.08, 1.21) | 0.13 | 2.28 | 5.11E-06 | 0.005 |
| Food | Raw carrots | −0.09 | 0.02 | 0.91 | (0.87, 0.95) | 0.06 | 1.1 | 6.33E-06 | 0.005 |
| Nutrient | Total saturated fat | 0.15 | 0.03 | 1.16 | (1.08, 1.24) | 0.78 | 3.14 | 1.20E-05 | 0.007 |
| Nutrient | Hydroxyproline | 0.11 | 0.03 | 1.12 | (1.06, 1.17) | 0.25 | 1.64 | 1.26E-05 | 0.007 |
| Nutrient | Isorhamnetin | −0.09 | 0.02 | 0.91 | (0.87, 0.95) | 0.32 | 1.27 | 1.59E-05 | 0.007 |
| Food | Liquor | −0.08 | 0.02 | 0.92 | (0.89, 0.96) | 0.12 | 1.06 | 2.06E-05 | 0.009 |
| Nutrient | Carbohydrate from wholegrain | −0.09 | 0.02 | 0.91 | (0.87, 0.95) | 0.22 | 1.28 | 2.46E-05 | 0.010 |
| Nutrient | Cereal fiber | −0.1 | 0.02 | 0.91 | (0.87, 0.95) | 0.2 | 1.49 | 4.04E-05 | 0.012 |
| Food | Red wine | −0.09 | 0.02 | 0.91 | (0.87, 0.95) | 0.63 | 1.04 | 4.28E-05 | 0.012 |
| Nutrient | Trans 18:2 | 0.11 | 0.03 | 1.12 | (1.06, 1.18) | 0.19 | 1.83 | 4.48E-05 | 0.012 |
| Nutrient | Dietary tocopherols | −0.13 | 0.03 | 0.88 | (0.83, 0.94) | 0.75 | 2.78 | 5.71E-05 | 0.013 |
| Nutrient | Palmitic acid | 0.14 | 0.03 | 1.15 | (1.07, 1.23) | 0.93 | 3.37 | 7.01E-05 | 0.015 |
| Nutrient | Dietary folate | −0.11 | 0.03 | 0.9 | (0.85, 0.95) | 0.49 | 1.91 | 8.33E-05 | 0.016 |
| Food | Doughnuts | 0.07 | 0.02 | 1.08 | (1.04, 1.12) | 0.1 | 1.1 | 9.84E-05 | 0.017 |
| Nutrient | Beta-tocotrienol | −0.09 | 0.02 | 0.92 | (0.88, 0.96) | 0.43 | 1.31 | 1.07E-04 | 0.018 |
| Nutrient | Plant MUFA | −0.11 | 0.03 | 0.9 | (0.85, 0.95) | 0.8 | 2.05 | 1.22E-04 | 0.019 |
| Food | Hotdog | 0.07 | 0.02 | 1.07 | (1.04, 1.11) | 0.05 | 1.09 | 1.26E-04 | 0.019 |
| Food | White bread | 0.07 | 0.02 | 1.08 | (1.04, 1.12) | 0.08 | 1.12 | 1.61E-04 | 0.022 |
| Nutrient | Natural germ | −0.08 | 0.02 | 0.92 | (0.88, 0.96) | 0.39 | 1.24 | 1.78E-04 | 0.022 |
| Nutrient | Apigenin | −0.08 | 0.02 | 0.92 | (0.89, 0.96) | 0.81 | 1.15 | 1.79E-04 | 0.022 |
| Nutrient | Beta-tocopherol | −0.1 | 0.03 | 0.91 | (0.86, 0.96) | 0.55 | 1.87 | 2.73E-04 | 0.028 |
| Nutrient | Natural bran | −0.08 | 0.02 | 0.92 | (0.89, 0.96) | 0.45 | 1.26 | 2.90E-04 | 0.028 |
| Nutrient | Supplemental selenium | −0.08 | 0.02 | 0.92 | (0.88, 0.96) | 0.89 | 1.38 | 4.01E-04 | 0.034 |
| Food | Apple juice or cider | 0.07 | 0.02 | 1.07 | (1.03, 1.11) | 0.95 | 1.07 | 4.49E-04 | 0.036 |
| Nutrient | Dietary manganese | −0.09 | 0.03 | 0.91 | (0.86, 0.96) | 0.08 | 2 | 4.82E-04 | 0.037 |
| Food | Peanuts | −0.07 | 0.02 | 0.93 | (0.89, 0.97) | 0.09 | 1.08 | 4.92E-04 | 0.037 |
| Nutrient | Alpha-tocotrienol | −0.08 | 0.02 | 0.92 | (0.88, 0.96) | 0.43 | 1.51 | 5.37E-04 | 0.037 |
| Nutrient | Myristic acid | 0.09 | 0.03 | 1.1 | (1.04, 1.15) | 0.59 | 1.99 | 5.44E-04 | 0.037 |
| Nutrient | Cholesterol | 0.09 | 0.03 | 1.1 | (1.04, 1.16) | 0.43 | 2 | 6.09E-04 | 0.039 |
| Nutrient | Supplemental or fortified folic acid | −0.08 | 0.02 | 0.92 | (0.88, 0.97) | 0.53 | 1.55 | 6.63E-04 | 0.040 |
| Food | All processed meats | 0.07 | 0.02 | 1.07 | (1.03, 1.11) | 0.23 | 1.14 | 6.73E-04 | 0.040 |
| Nutrient | Trans 18:1 | 0.09 | 0.03 | 1.09 | (1.04, 1.15) | 0.62 | 1.93 | 6.94E-04 | 0.040 |
| Nutrient | Total manganese | −0.08 | 0.02 | 0.92 | (0.88, 0.97) | 0.99 | 1.59 | 8.72E-04 | 0.046 |
| Food | Hamburger | 0.07 | 0.02 | 1.07 | (1.03, 1.11) | 0.09 | 1.18 | 9.53E-04 | 0.047 |
| Food | Beverages with sugar | 0.07 | 0.02 | 1.07 | (1.03, 1.11) | 0.86 | 1.13 | 9.70E-04 | 0.047 |
| Nutrient | Synthetic vitamin $B_6$ | −0.07 | 0.02 | 0.94 | (0.90, 0.97) | 0.66 | 1.07 | 1.02E-03 | 0.047 |
| Food | Cold breakfast cereal | −0.07 | 0.02 | 0.94 | (0.90, 0.97) | 0.13 | 1.1 | 1.02E-03 | 0.047 |
| Food | Raisins or grapes | −0.07 | 0.02 | 0.93 | (0.90, 0.97) | 0.91 | 1.11 | 1.09E-03 | 0.048 |
| Nutrient | Heme iron | 0.08 | 0.02 | 1.08 | (1.03, 1.14) | 0.51 | 1.55 | 1.17E-03 | 0.050 |

The exposures that are statistically significant in association with CHD risk are listed. P values are associated with two-sided Wald tests.

expected, foods high in negatively associated nutrients are also related to lower CHD risk and foods containing positively associated nutrients are related to higher CHD risk.

In the network shown in Fig. 3b, each node has its own estimated HR. For each nutrient, we compare the estimated HR with the expected geometric hazard ratio $\langle HR_i^N \rangle_f$, determined by

Eq. (1),

$$\langle \mathrm{HR}_i^N \rangle_f = e^{\sum_{k=1}^{j} w'_k \beta_k^F} = \prod_{k=1}^{j} (\mathrm{HR}_k^F)^{w'_k}, \qquad (1)$$

where $w'_k$ is the normalized weight of the link connecting nutrient

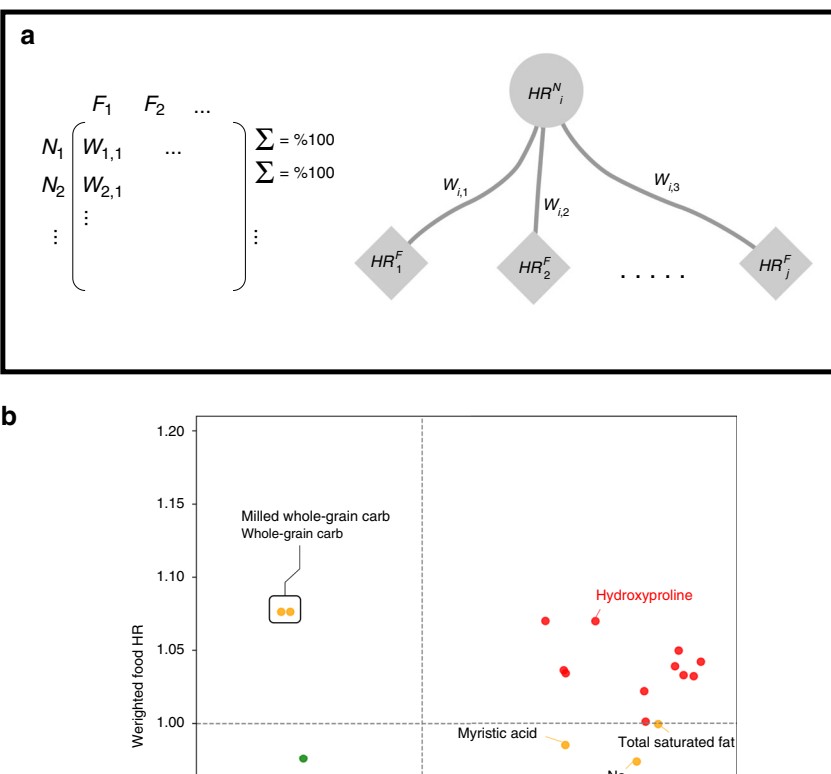

**Fig. 4 Comparison between Actual Nutrient HR and Weighted Food HR. a** The structure of the food composition table is shown on the left. A network is built based on a subset of the composition table where significant nutrients and significant food items are extracted. In the network on the right, the weight of a link between a nutrient and a food is proportional to the amount of a nutrient in each specific food item compared with all other food items. **b** Points in the top right corner and in the bottom left corner correspond to the nutrients whose actual HR is in the same direction as their weighted food HR. For whole-grain carbohydrates, milled whole-grain carbohydrates, total saturated fat, trans-fatty acid 18:2, sodium, and myristic acid, the actual HR has an opposite direction compared with the weighted food HR. Source data are provided in Source Data Figure 3-b and 4-b.xlsx.

$i$ to food $k$, $\beta_k^F$ is the estimated effect size of food $k$, $HR_k^F$ is the estimated hazard ratio of food $k$, and $HR_i^N \lessgtr \langle HR_i^N \rangle_f$ (Fig. 4a).

As we show in Fig. 4b, for the majority of nutrients, the actual nutrient HR has the same directionality as the weighed food HR, indicating that EWAS captures both important nutrients and their main drivers in the food system. Whole-grain carbohydrate and milled whole-grain carbohydrate were found to be negatively associated with CHD risk; however, among food items high in these two exposures, we only found doughnuts to relate with higher CHD risk statistically significantly. Consequently, the weighted food HR for these exposures has an opposite directional effect. Yet, the correlation analysis (Supplementary Fig. 3) shows that these two exposures have negative correlations with positively associated foods, such as white bread and processed meats, and positive correlations with negatively associated foods, such as cold breakfast cereal and raw carrots. Moreover, myristic acid, trans-18:2, sodium, and total saturated fat were found to be positively associated with higher CHD risk themselves, but the weighted food HR for them indicates an opposite direction. These nutrients are not only distributed among positively associated foods, but also negatively associated foods. Myristic acid is in both processed meats and yogurt, trans-fatty acid 18:2 is present

in salad/oil, vinegar dressing, and doughnuts, sodium is spread among salad/oil and vinegar dressing and processed meats, and total saturated fat is found in yogurt and processed meat. However, the consumption of these exposures is positively correlated with positively associated foods (Supplementary Fig. 3). The observed disparity indicates that for most of the nutrients, the structure of the food system determines the amount of nutrients in the diet, while for some other nutrients individual choices drive the nutrient amount in the diet. The signal determined by significant foods is a strong driver, but not always sufficient to capture exhaustively the nutrient associations with CHD. Even though some of the nutrients are not well captured by the food approximation (yellow points), with this approach we tend to correctly estimate the sign of the association, while underestimating the effect size. This observation indicates that solely looking at food items, one would underestimate the effect of those nutrients whose consumption is strongly determined by the behavioral aspect and not mainly by their average amount in food. Additionally, this observation can be partially explained by the higher resolution in the calculation of nutrient intake for breakfast cereals, margarine, and types of fats used for cooking and baking, for which we asked separate questions about their

type and brand that linked with extensive databases for detailed composition values.

**Comparison with the literature.** Applying the EWAS methodology to NHS data allowed us to identify 53 dietary exposures that show statistically significant association with CHD risk. Next, we discuss our results in the context of the previous literature, offering a direct validation of our findings, also helping detect novel knowledge, and to generate new hypotheses. We found that in most cases, EWAS strengthens the existing knowledge about the effect of diet on CHD, and in some cases, it sheds a light on exposures that have not been thoroughly studied in the literature.

Most of our statistically significant findings were in agreement with the previous literature, strengthening the prior findings and supporting the robustness of the EWAS platform. Among food items, we found that white wine, red wine, and liquor, but not beer, have a negative association with higher CHD risk, aligned with previous findings[35,36]. Despite recovering an inverse association between alcohol intake and CHD risk (HR 0.88; 95% CI: 0.84–0.91; $P$ value < $5 \times 10^{-11}$), it is worth noting that the overall level of alcohol consumption is not very high in this population study, and such inverse association has not been replicated in other quasi-experimental Mendelian randomization studies[37,38].

Moreover, we found salad/oil and vinegar dressing, yogurt, cold breakfast cereal, raw carrots, raisins or grapes, and peanuts have a negative association with higher CHD risk, in line with previous studies[39–44]. Moreover, we found that total processed meat consumption, hot dogs, apple juice or cider, beverages with sugar, and white bread have positive association with higher CHD risk, as previously shown by other studies[45–48].

Many compounds from lipid and fatty acid groups have been studied previously in relation to CHD risk. In line with previous studies[49–55], we found that higher consumption of cholesterol, trans-fatty acid 16:1, trans-fatty acid 18:1, trans-fatty acid 18:2, total saturated fat, animal monounsaturated fatty acids (MUFA), myristic acid, palmitoleic acid, palmitic acid, and stearic acid are associated with a higher risk of developing CHD ($P$ value < $1 \times 10^{-3}$). These dietary factors are mainly distributed among animal-based foods. By contrast, we found that plant MUFA can be protective against CHD development (HR 0.90; 95% CI: 0.85–0.95; $P$ value < $2 \times 10^{-4}$), which has also been shown by Zong et al.[53]. Plant MUFAs are abundant in salad/oil and vinegar dressing and peanuts, food items that we also found to be statistically significantly associated with lower CHD risk (salad/oil and vinegar dressing: HR 0.90; 95% CI: 0.87–0.94; $P$ value < $4 \times 10^{-7}$; peanuts: HR 0.93; 95% CI: 0.89–0.97; $P$ value < $5 \times 10^{-4}$).

Whole grains are composed of endosperm, germ, and bran, in contrast with milled whole grains in which only endosperm is retained. We found that higher consumption of carbohydrate from whole grains is associated with lower CHD risk, similar to the findings in ref. [56]. While the milling process removes several valuable compounds in whole grains, we interestingly detected a similar protective effect for carbohydrate from milled whole grains. In addition, we showed that both natural bran and added bran are negatively associated with CHD risk, consistent with previous studies[57,58]. We also documented a negative association for cereal fiber with CHD (HR 0.91; 95% CI: 0.87–0.95; $P$ value < $5 \times 10^{-5}$), in agreement with ref. [59]. One of the food groups that can be rich in natural bran and germ, added bran, and cereal fiber is cold breakfast cereal, which we also found to be negatively related to CHD (HR 0.94; 95% CI: 0.90–0.97; $P$ value < $2 \times 10^{-3}$).

While we rediscovered the negative association of manganese consumption with CHD risk (HR 0.92; 95% CI: 0.88–0.97; $P$ value < $1 \times 10^{-3}$)[60], we also found that higher supplemental selenium is associated with lower CHD risk (HR 0.92; 95% CI: 0.88–0.96; $P$ value < $5 \times 10^{-4}$). A specific cardiomyopathy responsive to selenium supplementation has been observed in domestic animals[61] and among Chinese persons with Keshan disease[62]. However, more recent studies found no association between selenium supplementation and primary prevention of cardiovascular disease (CVD), for which reason it is not recommended for CVD prevention[63,64]. Moreover, our observed positive association of sodium (HR 1.14; 95% CI: 1.08–1.21; $P$ value < $6 \times 10^{-6}$) is also consistent with prior studies[65].

Our findings indicate that higher consumption of dietary folate (HR 0.90; 95% CI: 0.85–0.95; $P$ value < $9 \times 10^{-5}$) and folic acid (HR 0.92; 95% CI: 0.88–0.97; $P$ value < $7 \times 10^{-4}$) are related to lower CHD risk, again aligned with previous findings[66,67]. While the beneficial effect of natural vitamin $B_6$ was previously documented[66], we interestingly found that increased synthetic vitamin $B_6$ consumption is related to lower CHD risk (HR 0.94; 95% CI: 0.90–0.97; $P$ value < $2 \times 10^{-3}$). Moreover, our results reveal that beta-tocopherol, total dietary tocopherol intake, alpha-tocotrienol, and beta-tocotrienol are negatively associated with CHD risk (HR < 0.92; $P$ value < $6 \times 10^{-4}$). Earlier studies reported similar effects regarding alpha-tocopherol, total tocopherol intake, and alpha-tocotrienol[68,69]. While the antioxidant and anti-inflammatory effects of some of the vitamin E isomers have been documented, little is known about the effect of beta-tocopherol and beta-tocotrienol on cardiovascular health. These two compounds are mainly found in whole-grain products and nuts. In the liver, beta-tocopherol undergoes omega-hydroxylation, oxidation, and beta-oxidation to generate 13′-hydroxychromanols/carboxychromanols, which have potential antioxidant properties[70,71]. This fact strengths our findings regarding the protective effect of beta-tocopherol against CHD. Moreover, beta-tocotrienol, in particular, was shown to be inversely related to the risk of type 2 diabetes mellitus[72], but has not been well-studied regarding its effects on the cardiovascular system. However, among different forms of tocotrienols, beta-tocotrienol has the highest antioxidant activity[73], in support of the protective effect against CHD that we observed in EWAS.

We found that heme iron (HR 1.08; 95% CI: 1.03–1.14; $P$ value < $2 \times 10^{-3}$) is statistically significantly associated with higher CHD risk. Similar effects were detected in ref. [74]. We also found isorhamnetin and apigenin (HR < 0.92; $P$ value < $2 \times 10^{-4}$) to be inversely associated with CHD risk, in line with refs. [75,76] (Supplementary Table 7). Moreover, in EWAS, we found that higher dietary hydroxyproline intake is associated with higher CHD risk (HR 1.12; 95% CI: 1.06-1.17; $P$ value < $2 \times 10^{-5}$). Hydroxyproline is a nonessential amino acid derivative and a major component of the protein collagen mainly found in animal-based food products, such as beef, chicken, and pork. Increased hydroxyproline levels in the urine and/or serum are normally associated with degradation of connective tissue and Marfan syndrome[77], and were also found to be related to Paget disease[78]. The decrease in various hydroxyproline fractions in aortic tissue of rabbits has been shown to be a risk factor for the progression of atherosclerosis[79]. Nonetheless, serum hydroxyproline is mainly associated with peptides released from the breakdown of collagen, and dietary hydroxyproline intake does not considerably affect serum hydroxyproline levels unless consumed in the form of gelatin[80]. While non-prescription hydroxyproline supplements are available as L-hydroxyproline and N-acetyl-L-hydroxyproline, there seems to be no evidence for the effectiveness of oral hydroxyproline supplements in the prevention or treatment of osteoarthritis, osteoporosis, rheumatoid arthritis, skin ulcers, sports injuries, and wrinkled skin, or in promoting muscle growth or weight loss[81]. The lack of

**Table 2 Validation in NHS II.**

| Type | Exposure | Effect size | SE (effect size) | Hazard ratio | 95% CI | P value of PH | VIF | P value |
|------|----------|-------------|------------------|--------------|--------|---------------|-----|---------|
| Nutrient | Cereal fiber | −0.28 | 0.06 | 0.75 | (0.68, 0.84) | 0.7 | 1.62 | 2.70E-07 |
| Nutrient | Total manganese | −0.28 | 0.06 | 0.76 | (0.68, 0.85) | 0.32 | 1.68 | 5.82E-07 |
| Nutrient | Alcohol | −0.21 | 0.04 | 0.81 | (0.74, 0.88) | 0.3 | 1.12 | 1.06E-06 |
| Nutrient | Beta-tocotrienol | −0.25 | 0.05 | 0.78 | (0.70, 0.86) | 0.56 | 1.39 | 1.92E-06 |
| Nutrient | Alpha-tocotrienol | −0.25 | 0.05 | 0.78 | (0.70, 0.87) | 0.84 | 1.52 | 3.64E-06 |
| Nutrient | Dietary manganese | −0.28 | 0.06 | 0.76 | (0.68, 0.85) | 0.74 | 1.96 | 3.72E-06 |
| Nutrient | Added bran from wheat, rice, etc. | −0.21 | 0.05 | 0.81 | (0.74, 0.89) | 0.79 | 1.13 | 5.86E-06 |
| Food | Cold breakfast cereal | −0.2 | 0.05 | 0.82 | (0.74, 0.89) | 0.16 | 1.11 | 1.18E-05 |
| Nutrient | Dietary folate | −0.25 | 0.06 | 0.78 | (0.69, 0.88) | 0.08 | 2.03 | 3.50E-05 |
| Nutrient | Natural bran | −0.2 | 0.05 | 0.82 | (0.75, 0.90) | 0.73 | 1.25 | 3.62E-05 |
| Nutrient | Carb from milled wholegrain | −0.2 | 0.05 | 0.82 | (0.75, 0.90) | 0.69 | 1.26 | 3.90E-05 |
| Nutrient | Stearic acid | 0.28 | 0.07 | 1.32 | (1.15, 1.51) | 0.45 | 2.64 | 6.26E-05 |
| Nutrient | Carb from wholegrain | −0.19 | 0.05 | 0.83 | (0.75, 0.91) | 1 | 1.28 | 6.47E-05 |
| Food | Salad/oil and vinegar dressing | −0.16 | 0.05 | 0.85 | (0.78, 0.93) | 0.15 | 1.17 | 3.41E-04 |
| Food | Raw carrots | −0.17 | 0.05 | 0.84 | (0.77, 0.93) | 0.93 | 1.14 | 4.02E-04 |
| Food | Red wine | −0.18 | 0.05 | 0.84 | (0.76, 0.93) | 0.72 | 1.07 | 7.00E-04 |
| Nutrient | Phytate | −0.21 | 0.06 | 0.81 | (0.71, 0.92) | 0.34 | 2.18 | 9.66E-04 |
| Nutrient | Beta-tocopherol | −0.21 | 0.06 | 0.81 | (0.71, 0.92) | 0.9 | 2.08 | 1.01E-03 |
| Nutrient | Apigenin | −0.15 | 0.04 | 0.86 | (0.79, 0.94) | 0.86 | 1.2 | 1.07E-03 |
| Nutrient | Supplemental or fortified folic acid | −0.17 | 0.05 | 0.85 | (0.77, 0.94) | 0.03 | 1.41 | 1.38E-03 |
| Nutrient | Discretionary solid fat | 0.21 | 0.07 | 1.23 | (1.08, 1.41) | 0.72 | 2.46 | 1.49E-03 |
| Nutrient | Natural germ | −0.14 | 0.05 | 0.87 | (0.79, 0.95) | 0.18 | 1.27 | 2.42E-03 |
| Nutrient | Total saturated fat | 0.22 | 0.07 | 1.25 | (1.08, 1.44) | 0.65 | 3.04 | 2.88E-03 |
| Nutrient | Trans 16:1 | 0.17 | 0.06 | 1.19 | (1.06, 1.33) | 0.47 | 2.06 | 3.27E-03 |
| Nutrient | Palmitic acid | 0.23 | 0.08 | 1.25 | (1.08, 1.46) | 0.94 | 3.31 | 3.47E-03 |
| Food | Beverages with sugar | 0.12 | 0.04 | 1.12 | (1.04, 1.22) | 0.29 | 1.11 | 4.38E-03 |
| Food | White wine | −0.14 | 0.05 | 0.87 | (0.79, 0.96) | 0.83 | 1.07 | 4.40E-03 |
| Nutrient | Synthetic vitamin $B_6$ | −0.13 | 0.05 | 0.88 | (0.80, 0.96) | 0.16 | 1.08 | 4.92E-03 |
| Nutrient | Trans 18:1 | 0.15 | 0.06 | 1.16 | (1.04, 1.29) | 0.17 | 1.71 | 7.17E-03 |
| Nutrient | Supplemental selenium | −0.12 | 0.05 | 0.89 | (0.81, 0.97) | 0.09 | 1.28 | 7.41E-03 |
| Nutrient | Palmitoleic acid | 0.16 | 0.06 | 1.18 | (1.04, 1.33) | 0.77 | 2.19 | 9.24E-03 |
| Nutrient | Animal fat | 0.16 | 0.06 | 1.18 | (1.04, 1.33) | 0.97 | 2.23 | 1.03E-02 |
| Nutrient | Animal MUFA | 0.15 | 0.06 | 1.16 | (1.03, 1.31) | 0.67 | 2.03 | 1.18E-02 |
| Food | Hotdog | 0.12 | 0.05 | 1.12 | (1.03, 1.23) | 0.75 | 1.1 | 1.18E-02 |
| Food | Raisins or grapes | −0.12 | 0.05 | 0.89 | (0.80, 0.98) | 0.28 | 1.08 | 1.90E-02 |
| Nutrient | Hydroxyproline | 0.12 | 0.05 | 1.12 | (1.01, 1.25) | 0.15 | 1.55 | 3.22E-02 |
| Food | Yogurt | −0.09 | 0.05 | 0.91 | (0.83, 1.00) | 0.14 | 1.1 | 3.95E-02 |

From 53 statistically significant exposures found in NHS, 37 were validated in NHS II. P values are associated with two-sided Wald tests.

effectiveness of dietary hydroxyproline is probably a consequence of its failure to be incorporated into collagen: only proline is bioavailable for this purpose. Proline only becomes hydroxylated during a later stage of collagen formation in order to facilitate the strengthening of the collagen helix. Once a collagen helix forms, it does so irreversibly in mammals[82]. The ambiguous role of dietary hydroxyproline, along with the positive association with higher CHD risk that we observed in EWAS, emphasizes the need for exploring the metabolic role that this amino acid plays in health and disease. A potential mechanism that can explain the positive association between hydroxyproline consumption and CHD risk is its metabolic reaction with succinate and $CO_2$, producing 2-oxoglutarate. Chen et al.[83] showed that increased serum 2-oxoglutarate is associated with high myocardial energy expenditure and poor prognosis in chronic heart failure patients.

**Validation in NHS II**. In the second phase of our study, we used NHS II in order to validate statistically significant associations that we found in the original NHS. During 20 years of follow-up in NHS II, 90,861 participants were followed and 604 CHD incidents were documented (Supplementary Methods 3.1). We examined the relationship between 53 exposures found in NHS with CHD risk and deemed an exposure tentatively validated if it had achieved a false discovery rate (FDR) <0.05 significance in

NHS and achieved nominal statistical significance in NHS II (P value < 0.05). Tentatively validated exposures in NHS II had the same directional association with CHD risk as in the original NHS (Table 2). A list of validated and non-validated associations is provided in Supplementary Table 5.

## Discussion

Our analysis of the dietary determinants of CHD has several limitations based on the nature of the data we used in our analysis. First, the study subjects are only women with a specific occupation (nurses), which restricts the generalizability of the findings to populations comprising males, as well as to more heterogeneous occupational groups and socio-economic backgrounds (Supplementary Figs. 4 and 5). Second, the present study only focused on the effect of dietary factors on CHD risk. Even though diet is an important part of an individual's environmental exposure, it does not cover the entire *exposome*[84]. The environment also includes persistent organic pollutants, plastic-associated chemicals, bacterial and viral infections, air quality, stress, and social network effects[85], as well as the endogenous microbiome. Moreover, our diet is not limited to nutrient content; it also carries food additives and other chemicals added during the packaging process, which are absent in the food composition databases, and hence are not included in this

analysis. Including these chemicals is necessary for a more comprehensive picture of the effect of the diet on health[86,87]. The only way to overcome these limitations is to include more environmental factors, offering a more comprehensive understanding of the global environment's effects on health and disease. Moreover, the FFQ used in NHS covers a number of food items that includes the large majority of those consumed by Americans, but the diet of some participants may not be completely represented. Third, while we included the confounding variables that were previously used in other studies on CHD relying on NHS data, our study remains limited to the common confounding variables usually considered when exploring diet–CHD associations. Hence, residual confounding by unmeasured variables cannot be excluded. The obtained results do not unveil causal effects, but, rather, help us generate new hypotheses, which need to be examined in more detail in these and other prospective cohorts and experimental studies. We must also investigate carefully and mechanistically the influence of these dietary factors on human metabolism, exposures that require detailed measurements in terms of dietary bioavailability, hence they can serve as targets for further investigation for mechanism-based analysis.

Our overall goal was to apply GWAS-like analytical approaches to study the dietary determinants of CHD. The methodology allowed us to explore both food items and nutrients, offering a more comprehensive picture of the effect of diet on CHD and helping us visualize the obtained relationships using network tools. Our study not only reproduced the prior knowledge in the diet–CHD domains, but also led to novel associations. While some of the previous EWAS studies failed to achieve adequate statistical power in association detection[19,20], our positive results suggest that these failures were often related to the cohort size and the absence of repeated longitudinal dietary assessments. Indeed, our use of a large longitudinal dataset with a long follow-up period and a sufficient number of subjects helped us achieve sufficient statistical power to detect even relatively small effect sizes. However, enabling a wide-association study to investigate environmental factors requires careful consideration in designing cohort studies, and detailed, comprehensive exposure assessment methods to ensure that the effect of the environment is fully captured. While selectively testing and reporting one or a few associations has been argued to be a source of biased results and false positives[88,89], there is clearly a role for testing specific etiologic hypotheses as this allows greater statistical power and a more detailed examination of an exposure–outcome relationship. An environment-wide association study is a complementary approach that allows us to rank the associations and report transparently both significant and non-significant associations. It also allowed us to generate new hypotheses that can be further investigated in single-association studies and mechanism-based studies. It is worth mentioning that until recently NIH was unlikely to fund research proposals without well-developed hypotheses, restricting the possibility of conducting wide-association studies and this analysis was only possible because of the accrual of over 30 years of follow-up in this large cohort.

In the present study, we explored the effect of only 374 dietary exposures. Yet, when it comes to the chemical composition of the food we consume, these nutritional components represent only a tiny fraction of the thousands of distinct definable biochemicals that have been identified in foods[90]. While many of these chemicals have well-documented or potential implications for health, they remain largely unquantified in any systematic fashion across different individual foods. Their invisibility to experimental, clinical, epidemiological, and demographic studies—turning them into the virtual *dark matter* of nutrition research—represents a roadblock toward a better, more consistent, more reproducible understanding of how diet affects health[91,92]. In the high-resolution diet description space, the conventional single-association approach is even more impractical and lacks scalability. The EWAS methodology, however, would be able to test higher order of magnitude of dietary compounds, to identify significant associations with a disease of interest or with a prescription for health.

## Methods

**Knowledge graph**. To create the knowledge graph, we firstly identified in PubMed all papers that have NHS or Health Professionals Follow-up Study in the title or abstract, along with papers co-authored by the main PIs of the NHS. We manually filtered papers that studied the association between dietary exposures and cardiovascular complications, such as coronary heart disease, stroke, and hypertension. Since not all papers are indexed on PubMed, we searched the web using the same criteria to find the remaining papers. Next, we manually examined the abstracts of the obtained papers and extracted the exposure–phenotype relations, the associations found, the effect size, and other related information (Table 3). More than one association might be studied in a paper. Overall, we found 292 studied associations documented in 91 papers, altogether 124 negative and 45 positive associations were documented in relation to cardiovascular complications. In the remaining cases, there was no significant association between an exposure and a phenotype of study. The obtained data are shown in a knowledge graph (Fig. 1) where each link represents an association. The space of studied exposures is rather heterogeneous and is often driven by the researcher's interests and experience. For example, the effects of some nutrients were studied with respect to replacement with other nutrients, such as the effect of replacing *trans*-fat with MUFAs. In some cases, the intake ratio of two nutrients, for example, the ratio of polyunsaturated fat to *trans*-fat, or of two food items was examined. Moreover, according to the exposure of interest, the set of adjusting variables used to account for confounding effects varied from one study to another. The raw data and code used for constructing the knowledge graph are available at ref. [93].

**Population**. Using the year 1986 as the baseline because the dietary questionnaires have been unchanged since, we followed women who were healthy and free of chronic diseases up to 2014. In the baseline year, participants with a history of CVD, diabetes mellitus, and cancer were excluded. We also excluded women whose demographic data were missing, whose reported average energy intake was <600 or >3500 kcal/day, or left >70 questions in the FFQ unanswered[10,94–97]. Participants received one questionnaire every 2 years to report their medical data and one questionnaire every 4 years to document their dietary data. At any point within the follow-up period, if a participant reported development of non-fatal myocardial infarction (MI) or fatal CHD[98–100], she will be classified as a case, with no further update of her dietary records. If she developed other diseases, such as diabetes mellitus or cancer, she would still be classified as a non-case, with no further update of her dietary records (Supplementary Methods 3.1 and Supplementary Fig. 1). These exclusion criteria were chosen by virtue of minimizing reverse causation bias and reducing the impact of measurement errors and missing data. In total, we included 62,811 subjects in the analysis, representing 2774 cases (4%) and 60,037 controls (96%).

**Ascertainment of diet**. NHS uses an internally designed FFQ, with documented reproducibility and validity[10,101,102]. The FFQ has been regularly updated to adapt to changes in the food market and to capture additional food items[103]. For each food item, the FFQ specified a commonly used unit or portion size, asking each subject how often, on average, she had consumed that quantity during the past year[104]. Nine responses were possible, ranging from "almost never" to "six or more times per day." We converted the frequency responses to the number of servings per day for each food item. We calculated daily intake of nutrients by multiplying the frequency of consumption of each item by its nutrient content and summing the nutrient contributions of all foods on the basis of Harvard University Food Composition Database derived from US Department of Agriculture sources[105] and other resources, including published reports, data from manufacturers, and in-house analyses of fatty acid composition[106]. We looked into several food items more closely. For example, we used an algorithm designed by Jacobs et al.[107] to classify breakfast cereals into wholegrain and refined grain. We also collected detailed information on the type of fat or oil used in food preparation and brand or type of margarines to calculate the fatty acid consumption.

**Ascertainment of CHD**. We ascertained incident cases of CHD (non-fatal MI or fatal CHD) that occurred after the return of the 1986 questionnaire but before June 1, 2014. Physicians, unaware of the self-reported risk factor status, systematically reviewed the medical records of those who reported having an MI on each biennial questionnaire. MI was classified as confirmed if the World Health Organization criteria, that is, symptoms, electrocardiographic changes, or elevated cardiac enzyme concentrations, were met[108]. Fatal CHD was confirmed by either hospital records or through an autopsy if CHD was listed as the cause of death on the death certificate, if it was listed as an underlying and most plausible cause of death, or if evidence of previous CHD was available. Deaths were identified from state vital

**Table 3 Knowledge graph data preparation.**

| PubMed ID | Year | Title | Abstract | Exposure | Phenotype | Gender | Association | Model | Effect size |
|---|---|---|---|---|---|---|---|---|---|
| 29529162 | 2018 | Carbohydrate quality and quantity and risk of coronary heart disease among us women and men | ...We aimed to assess the relation between various measures of carbohydrate quality and incident CHD. Data on diet and lifestyle behaviors were prospectively collected on 75,020 women and 42,865 men participating in the Nurses' Health Study (NHS) and the Health Professionals Follow-Up Study (HPFS) starting in 1984 and 1986, respectively, and every 2–4 years thereafter until 2012... In models adjusted for age, lifestyle behaviors, and dietary variables, the highest quintile of carbohydrate intake was not associated with incident CHD (pooled RR = 1.04; 95% CI: 0.96, 1.14; P trend = 0.31). Total fiber intake was not associated with risk of CHD (pooled RR = 0.94; 95% CI: 0.85, 1.03; P trend = 0.72), while cereal fiber was associated with a lower risk for incident CHD (pooled RR = 0.80; 95% CI: 0.74, 0.87; P trend <0.0001). In fully adjusted models, the carbohydrate-to-total fiber ratio was not associated with incident CHD (pooled RR = 1.04; 95% CI: 0.96, 113; P trend = 0.46). However, the carbohydrate-to-cereal fiber ratio and the starch-to-cereal fiber ratio were associated with an increased risk for incident CHD (pooled RR = 1.20; 95% CI: 1.11, 1.29; P trend <0.0001, and pooled RR = 1.17; 95% CI: 1.09, 1.27; P trend <0.0001)...[59] | Total fiber | CHD | Both | No association | Cox | - |
|  |  |  |  | Cereal fiber | CHD | Both | Negative | Cox | 0.8 |
|  |  |  |  | Ratio of carb to total fiber | CHD | Both | No association | Cox | - |
|  |  |  |  | Ratio of carb to cereal fiber | CHD | Both | Positive | Cox | 1.2 |

statistics records and the National Death Index, or were reported by the families and the postal system[47].

**Statistical analysis**. Figure 2a shows a brief snapshot of the statistical approaches used in this paper. We used the extended Cox model for time-dependent variables to associate each exposure with the time to occurrence of CHD. The underlying time for the Cox model is the time on study for each participant. We used the cumulative average (Supplementary Methods 3.2) of the food intakes from baseline to the start of each 2-year follow-up interval, which represents the long-term habitual intake and reduces random within-person variation[10,109–116] (for analyses in which the time-dose effects are taken into account see Supplementary Table 4 and Supplementary Fig. 6). We adjusted the analyses for potential risk factors and confounders, including age (Supplementary Fig. 2), BMI, physical activity, and total caloric intake as continuous covariates; and ethnicity, smoking status, multivitamin use, vitamin E supplement use, post-menopausal hormone use, aspirin use, high blood pressure[117], elevated cholesterol[118], and family history of MI and high blood pressure as categorical variables. We selected the set of confounding variables based on their potential effects on both exposures and the outcome. Dietary exposures entered the analysis as continuous variables. We used Box–Cox transformation to stabilize the variance and improve the validity of measures of association. Later, the exposures were z-transformed in order to compare the effect sizes from many regressions.

To examine the validity of the EWAS results, we assessed the proportionality assumption for each test. We also tested whether there was severe multicollinearity among the variables in each test by calculating the VIF, which can potentially make effect size estimates unstable, reduce or eliminate statistical power, and cause the coefficients to switch signs[119]. Ultimately, to control for type I error due to multiple hypotheses testing, we calculated the FDR, the estimated proportion of false discoveries made versus the number of real discoveries at a given significance level[120] (Supplementary Methods 3.3). To estimate the number of false discoveries, we created a null distribution of Cox model P values by randomly shuffling the CHD status 1000 times and recomputing the P values. Accordingly, we estimated the FDR to be the ratio of the proportion of results that were called significant at a given level α in the null distribution and the proportion of results called significant from real tests. Since in the FDR estimation, we utilize the data themselves, we naturally consider the correlated structure of the data, given the intrinsic dependencies among dietary factors[121]. Since the confounding effect of the adjusting covariates on CHD risk exists, certain subjects have greater odds of developing CHD. Therefore, we maintained the confounding role of the adjusting covariates in each permuted dataset while the association between the exposure and CHD has been eliminated[122]. We used Cox regression to estimate the odds of developing CHD as a function of adjusting covariates. Next, we permuted the CHD cases among the subjects as taking a random sample from a biased pool. Furthermore, we re-ran the analyses and calculated the null P values. Repeating this procedure 1000 times, we measured the FDR as the ratio of the proportion of results that were called significant at a given level α in the null distribution to a proportion of results called significant from our real tests.

**Study protocol**. The study protocol was approved by the institutional review board (IRB) of the Brigham and Women's Hospital, and the IRB allowed participants' completion of questionnaires to be considered as implied consent. Written informed consent was obtained from participants to release medical records documenting the incidence of coronary heart disease.

**Reporting summary**. Further information on research design is available in the Nature Research Reporting Summary linked to this article.

## Data availability

The authors declare that all data supporting the findings of this study are available upon request to Nurses' Health Study (NHS) and when the request for data access is approved. Access is restricted due to participant confidentiality and privacy concerns. Individuals who want to request access to NHS data must first submit an online request form. If the project is approved, completion of a data use agreement, completion of CITI training demonstrating ethical training in using human subjects' data, and a small provision of funds to support the computing system will also be required. Further information including the procedures to obtain and access data from the NHS is described in ref. [123] (contact email: nhsaccess@channing.harvard.edu). The Food Frequency Questionnaires used in NHS are available in ref. [104]. Harvard University Food Composition Database can be accessed in ref. [106]. Source data are provided with this paper.

## Code availability

The programming materials are available on GitHub and Zenodo platforms[93]. R version 3.4.0 was used for data analysis and Python version 2.7.16 and MATLAB 2019a were used for data visualizations.

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

## Acknowledgements

We were supported by grants from NIH (grants P01 HL132825, UM1 CA186107, P01 CA87969, R01 CA49449, R01 HL034594, R01 HL088521, UM1 CA176726, R01 CA67262, U54 HL119145, U01 HG007690, and P50 GM107618) and AHA (grants 151708, 414110-68953, and D700382). A.-L.B. was supported by NIH 1P01HL132825, Rockefeller Foundation 2109 FOD 026, and the European Union's Horizon 2020 research and innovation programme under grant agreement No 810115 - DYNASNET.

## Author contributions

S.M. performed data query and integration, statistical modeling, network analysis, and programming and contributed to writing the manuscript. G.M. contributed to network analysis, statistical modeling, programming, and writing the manuscript. Y.L. contributed to data query and programming. J.L. contributed to interpreting the results and writing the manuscript. W.C.W. contributed to data collection, analyzing the results, and writing the manuscript. A.-L.B. contributed to the conceptual design of the study and writing the manuscript.

## Competing interests

A.-L.B. is founder of Nomix and Foodome, and J.L. and A.-L.B. are founders of Scipher Medicine, companies that explore the use of network-based tools in health. The remaining authors declare no competing interests.
