## [Peer Review File · Nature Communications]

Reviewers' comments:

Reviewer #1 (Remarks to the Author):

The authors have applied a new approach (an Environment-Wide Association Study -EWAS-) to data from Nurses' Health Study (NHS) to explore the effects of many foods and nutrients on Coronary Heart Disease (CHD) risk. The authors have included 62,811 women in the analysis (2,774 cases -4%- and 60,037 controls -96%-). With this technique, the authors identified only 16 food items and 37 nutrients that show statistically significant association with CHD risk. The paper is well-written and very comprehensive, and results and conclusions are very interesting, but the manuscript may be improved if the authors take into account the following:

1. Dietary patterns: Traditionally, nutrition research has focused on single nutrients or specific foods, as the current study, although individuals do not consume nutrients or foods in isolation. Several nutritional epidemiologic studies have shifted to dietary pattern analysis, which describes the overall diet (Am J Clin Nutr.2015;101:899–900). Overall diet capture food groups, nutrients and, more important, their combination and variety; Thus, dietary pattern may allow a closer relationship analysis with health outcomes. The authors should include a paragraph in the introduction on this issue, and explain the usefulness of their food and nutrient analysis in front other approaches such as dietary pattern analyses.
2. Polyphenols: The authors have performed a deep analysis on the relationship between several nutrients and CHD outcomes, including several subclasses of polyphenols (Tables). However, the role of these compounds on CHD has not been included in the results and consequently in the discussion. This reviewer believes that polyphenols merit a greater attention (Biochem Pharmacol.2018;156:186-195).
3. Potassium and Magnesium: Both ions have important effects on health and also merit to be considered.
4. Beer: In addition to white and red wines, beer also merits some attention in the manuscript (at least a sentence).

Reviewer #2 (Remarks to the Author):

The manuscript by Milanlouei and Menichetti et al. explores how dietary factors associated with heart disease risks can be systematically analyzed using a framework inspired by Genome-Wide Association Studies. The authors dub this Environment-Wide Association Study (EWAS). The results

are further analyzed by breaking down food items into nutrients. I think the paper is interesting, however, there are many unclear technical issues as well as issues in relation to disease definitions. It is positive that the authors validate their findings in the NHS-II cohort, it would strengthen the paper further if validation in the UK Biobank cohort was added, this cohort of 500k individuals includes both men and women.

I issues with the manuscript as is, which made it difficult to ascertain whether the associations are valid.

Introduction

The introduction is well written, however, it could mention two of the larger prospective cohorts that also record diet (UK Biobank, Kadori Biobank). Only very little is mentioned about the underlying genetics of CVD.

Methods

The methods section appears to be incomplete in several ways:

1. The paper lacks a proper description of the two cohorts used. Number of participants, number of participants lost at each follow-up, average number of follow-up years etc. It was not until the Discussion that it was mentioned that the cohort consisted of women only.
2. Ascertainment of CHD (p. 18): The definition of cases (acute myocardial infarction (MI) and fatal CHD) and non-cases: Is it understood correctly that cases survive the acute phase? Also, the annotation does not seem to be fully aligned as you write nonfatal MI in the Results section (p. 4).
3. It is unfortunate that the all details of the underlying code are not made available (see below).
4. Individuals with >70 patients unanswered questions removed. This is unfortunate and is likely to bias the analysis unless data is missing completely at random. See e.g. <https://www.ncbi.nlm.nih.gov/pmc/articles/PMC1839993/>. With the Cox regression model, it should be simple to use e.g. multiple imputation as implemented in mice to obtain an unbiased estimate. Furthermore, is this to be understood as you also removed individuals with incomplete follow-up? This would also be problematic, and bias the analysis a lot.
5. Why censor individuals developing diabetes/cancer? Why not just include variables for diabetes/cancer in the model?
6. In Supplementary Figure 2, this is indicated to be a case-control study. This is not a case-control study, it is a cohort study. There are no controls, only individuals not developing the outcome.

7. Population (p. 17 + 18 and Figure SI. 1). Handling of cases and noncases. It reads as if cases must develop acute myocardial infarction or fatal CHD as their first disease. Otherwise the NHS participant is classified as a non-case. What is the underlying rationale here? Is it primarily a statistical motivation or a phenotypic? In the latter case, the workflow is a bit misleading unless you also present the counts on how many NHS participants develop diabetes and the acute myocardial infarction, for instance. A priori one would think they ought to be taken into account.

8. Please include a description of how many women were removed at each step.

9. "our own analyses of fatty acid composition" - Please describe this. Furthermore, including the compiled list of nutrients by food item as a supplementary file would be very useful for future studies.

11. "Ascertainment of CHD" - can you please confirm if this was done as part of the current study, or just in general for the Nurses Health Study?

I also have some concerns with the statistical analysis:

1. When using the extended Cox model, did you include a baseline risk for each participant to consider having a different number of data points for each individual?

2. Please indicate the underlying time for the Cox model

3. Age was modeled as a linear covariate. What is the rationale behind this? Why not include e.g. a spline to take into account non-linear patterns in the association between CVD and age?

4. Not being fully familiar with NHS, have hormone contraception been recorded at other time points? This could also be very relevant to adjust for.

5. When using the cumulative average of food intakes, did the authors attempt to do any outlier detection? Or are these results presented somewhere?

6. "We selected the set of adjusting variables based on their contribution in explaining the variability in CHD risk" - What exactly does this mean? Is this analysis part of this work?

7. How was the proportional hazard assumption verified?

8. The authors cite a study used to correct for confounders in case-control studies. However, this is not a case-control study. It is a cohort study. Case-control studies are typically enriched for cases, compared to cohort studies. Can the authors please justify how a case-control procedure generalises to a cohort study, and if it introduces any bias? Also, 1,000 permutations seem very little considering the power of modern computers.

9. Instead of assuming each test is independent, which is it not, could the authors do e.g. a PCA decomposition of the individual x food matrix to approximate how many effective tests the authors are performing? See e.g.

<https://journals.plos.org/plosgenetics/article?id=10.1371/journal.pgen.1006711>. Also, please note

that the Bonferroni correction is not used in a GWAS setting as described. Instead, the number of individual tests, due to correlations in the genome, have been estimated.

10. Ascertainment of Diet. "Nine responses were possible, ranging from almost never" to six or more times per day." We converted the frequency responses to the number of servings per day for each food item." How did you transform the categorical variables to continuous variables? While the frequencies between 1-5 times/day are simple, the conversion for "almost never", is unclear, as 0? And six or more? Was it transformed to 6?

11. "We calculated daily intake of nutrients". How did you decide on the units for nutrient amounts? For that you need to specifically know the food intake amount.

12. The nutrient content of each food depends on many factors, of which the brand is important. In the cited paper's (reference 83) database one can search for a specific brand of food. How did you manage to obtain the nutrient information of unspecific foods, such as: apple juice, hot dog, etc... without searching for specific brands?

It is well known that different brands use a broad range of different food additives which are not part of the nutrient content but have the potential to induce adverse reactions in the organism, which in turn could be associated to development of CHD. Some examples are the food additives nitrates and nitrites which have been associated to carcinogenicity, and thyroid hormone disruption (Bouvard V, Loomis D, Guyton KZ, et al.; International Agency for Research on Cancer Monograph Working Group. Carcinogenicity of consumption of red and processed meat. *Lancet Oncol.* 2015;16(16):1599–1600

Tonacchera M, Pinchera A, Dimida A, et al. Relative potencies and additivity of perchlorate, thiocyanate, nitrate, and iodide on the inhibition of radioactive iodide uptake by the human sodium iodide symporter. *Thyroid.* 2004;14(12):1012–1019)

The food additives should also be added to the study, in order to fully cover the spectrum of exposure. Besides, these food additives could be the ones behind the underlying adverse effects.

Furthermore, there is several indirect food additives, which are chemicals that are not directly added to the food but are put in contact with it among others through the packaging process. These include bisphenol, phthalates, etc. Although they cannot be taken into account fully, since they are widely unknown, it is important to mention them as a limitation of these type of study.

13. "our own analyses of fatty acid composition". What does this mean? Where do you describe this analysis?

14. The authors fail as far as I can see to explain the time-dose use. The authors used a cumulative average food over a 2 year interval. This approximation fails to handle dose of exposure changes over time, which often are critical if one wants to understand the effects of the nutrients in disease development. The intake of a nutrient does not necessarily produce an adverse outcome, unless there is a continued specific dose exposure. For example, alcohol has been associated in this study with a protective effect. However, in high doses, alcohol has previously been associated with disease development (Alcohol consumption and risk of stroke: a meta-analysis, Reynolds K, Lewis B, Nolen JD, Kinney GL, Sathya B, He *JAMA.* 2003 Feb 5; 289(5):579-88.) .

15. There seems to be several issues with the code, including the lack of a data file that could be used to perform a test run. The authors also fail to define the format of the input data as far as I could see. The rest of the code seems well formulated and commented apart from these aspects: 1) The authors fail to explain what the first function does, "addToBase"; 2) Re "na.locf", this step adds the previous value to all the following rows which contain NA. Without a data table one cannot assess whether this is a biased manual imputation, or an imputation that correlates to what the authors have defined in the Methods section of the paper; 3) "temp_var <- c(covar, 'calorconv')". What does "calorconv" refer to?

Results

1. Figure 1: The knowledge graph (p. 5). The difference between the diamonds seems somewhat unclear. E.g. What is the difference between coronary heart disease (CHD) and coronary artery disease? And what phenotype does cardiovascular disease (CVD) include that is not included in any other diamond. According to the Supplementary Information section 1 there were 124 negative and 45 positive associations documented in relation to heart-related complications. However, translation to the knowledge graph seems fuzzy.
2. Figure 2a: I do not understand this figure. Please provide a clearer explanation in the Statistical Analysis section instead. Again, Cox estimates hazard, not odds.
3. "The minimum statistical power for detecting the smallest absolute effect size was 0.59, which is considered to be a moderate to high level of power in clinical studies" - post-hoc power calculations, or "observed power" are not appropriate.
4. "Traditional epidemiological studies are limited to the detection of a single exposure (food or nutrient) in relation to CHD". That is absolutely not true. See e.g. <https://academic.oup.com/ije/article/36/3/600/652363>.
5. Please move the description of the network construction to the Methods section.
6. "Below, we discuss both non-validated and validated associations found by EWAS in the context of previous literature.". There is nothing below.

Conclusion

1. "Third, while we included only the adjusting variables with statistically significant effects on CHD risk rather than selecting them subjectively, our study remains limited to the common covariates usually considered when exploring diet-CHD associations". Again, what selection was made exactly?

2. "Indeed, our use of a large longitudinal dataset with a long follow-up period and a sufficient number of subjects helped us achieve sufficient statistical power to detect even relatively small effect sizes". Is this necessarily a strength? Very small effect sizes often have little or no real life value. It would be good to discuss how exactly the hazard ratios should be interpreted. E.g., if I drink one unit of alcohol, how much does that affect my risk? What about a carrot?

3. Conclusion and Future Work (p. 15 + 16). It is surprising that smoking is not included in the discussion. Is it considered part of "air quality" or is it left out for some – seemingly – less obvious reason?

4. The authors have only focused on CHD and have eliminated any other comorbidity from the analysis, which can be correlated to the development of this disease. The authors should perform a competing risk analysis in which there are several possible outcome events, in this case any of the diseases shown in Figure 1.

Supplementary

line 195-198. I do not understand how the approach utilises the structure in dietary information. As far as I can read, only the confounding variables are used.

Minor

spelling error: "hazard ratio for one stabdard deviation"

"We used Cox regression to estimate the odds of developin" - The Cox regression model estimates hazard ratios, not odds.

Reviewer #3 (Remarks to the Author):

This is an important and timely paper that aims to overcome the limitations of most literature on nutrition and health, in particular the single-item approach. They apply advanced statistical methods to disentangle the effects of nutrients and foods taking their inter-correlations into account. They also validate the results in a separate dataset.

As such, this is a nice and well-written paper, though the authors are aware it is only a first step in advancing the field:

- We know a limited number of components in food (the tip of the iceberg). I like the analogy with dark matter
- There is some degree of mismeasurement that implies misclassification
- Foods act via nutrients and other components, meaning that rather than a correlational analysis next steps could include mediation analysis (as they rightly point out, there is a many-to-many relationship between foods and their constituents)

These advancements could be left for next papers.

Main comments:

- I would have expected a pattern analysis, and in particular a visual representation of inter-relationships like in an “exposome globe”
- The example of Kolonel and papaya looks a little out of place, being only one example that might be due to chance
- Notice that in the paper by Merritt coffee was inversely associated with endometrial cancer
- In the introduction the main limitation of studies seems to be related to sample size, but there are many others: misclassification of food habits, intra-individual variation, temporal variation, lack of repeat measurements, etc

- the limited statistical power of previous studies is not absolute. EPIC and UK Biobank are large cohorts

- I am not completely persuaded by the analogy with GWAS. In GWAS one is interested in single gene variants, confounding is limited and the structure of the data is completely different from EWAS, being dictated by haplotypes

Reviewer #4 (Remarks to the Author):

This paper is using an EWAS approach to identify effects of diet on coronary heart disease in women (CHD) using the Nurses Health Study. Observational studies of the effects of diet on health are well-known to be difficult to interpret because of bias from confounding, i.e., it is difficult to separate the attributes of the individual from the effect of the exposure. To put it another way many, many factors may determine both diet and health, i.e., are common causes of diet and health, and these factors will bias the observed associations of diet with health unless they are fully accounted for in the analysis. Prime factors that might affect diet and health include all aspects of socio-economic position, including parental attributes, household income, wealth, familial status and level of education, which are impossible to measure comprehensively and precisely. To give unbiased estimates of the effects of diet on health the analysis must be fully adjusted for all factors that could bias the observed associations, i.e., all confounders of the association of diet on CHD, where confounders are defined as common causes of exposure and outcome.[1] Given confounding is a causal concept,[1] confounders cannot be identified reliably from observational data.

Each association in this study should be adjusted for all factors that could confounder the association of diet with health, i.e., common causes of diet and CHD. In this study the concept of confounding is not used, and instead the analyses are adjusted for covariates (figure 2) identified from the data (methods). The covariates included in the analysis do not include key confounders, such as all aspects of socio-economic position, but do include factors that are unlikely to be confounders, such as aspirin use, high blood pressure and elevated cholesterol. As such, each of the associations in the EWAS are highly likely to be biased by unmeasured confounding and potentially by mediators. The EWAS analysis of these associations is thorough and comprehensive, but it is not designed to be nor is capable of rectifying the fact that each association included is potentially biased, meaning that the findings cannot be interpreted.

Specific comments

Abstract

Please give a more nuanced interpretation of the results of dietary epidemiology

Please avoid the use of the word effect when describing observed associations

Please clarify the extent to which this study has replicated the effects of diet on health established from a systematic review of experimental or quasi experimental studies, such as randomized controlled trials or instrumental variable analyses.

Please make it clear that the integrity of the study depends on assuming no unmeasured confounding of the associations of dietary items with health, and list in the abstract the confounders considered so that the reader can make their own assessment of the validity of the findings.

Introduction

Please give a more nuanced assessment of the achievements of dietary epidemiology. Many vitamins and minerals identified from dietary epidemiology as protective were not validated in randomized controlled trials, and some even turned out to be harmful[2].

Please explain more clearly why EWAS overcomes problems of correlated exposures and unmeasured confounding.

Please give evidence for the statement “..we consistently recover prior knowledge about diet-disease relationships”, and clarify that this means recovering the results of randomized controlled trials, rather than replicating the previous observations from the Nurses Health Study.

Methods

Please use the up-to-date definition of a confounder

Results

Please amend the box in Figure 2a from “adjusting covariates” to “adjusting confounders” and amend the analysis accordingly throughout.

Comparison with the previous literature

This section should give a comparison with evidence from randomized controlled trials and quasi-experimental studies. There is little value in showing that EWAS replicates findings known to be uninterpretable due to bias.

Validation needs to be using data and designs which are not open to the same biases as this study, so observational analysis of Nurses Health Study II cannot provide validation.

Conclusion and Further Work

Please explain that this study is open to unmeasured confounding using an up-to-date definition of confounding with a more realistic assessment of the effects of unmeasured confounding.

Please consider whether the other major source of bias in observational studies, i.e., selection bias

In view of these limitations please do not interpret this study as a guide to the causal effect of diet, but instead contextualize it within the more recent evidence from randomized controlled trials and mendelian randomization studies.

1. Bareinboim E, Pearl J. Causal inference and the data-fusion problem. *Proceedings of the National Academy of Sciences of the United States of America*. 2016;113(27):7345-52. PubMed PMID: 27382148.

2. Lawlor DA, Davey Smith G, Kundu D, Bruckdorfer KR, Ebrahim S. Those confounded vitamins: what can we learn from the differences between observational versus randomised trial evidence? *Lancet (London, England)*. 2004;363(9422):1724-7. Epub 2004/05/26. doi: 10.1016/s0140-6736(04)16260-0. PubMed PMID: 15158637.

Statement on the Revision of ⟨NCOMMS-19-19797⟩

Based on the Referees' Report

Soodabeh Milanlouei Giulia Menichetti Yanping Li Joseph Loscalzo
Walter C Willett Albert-László Barabási

December 2, 2019

Comments by Reviewer #1

The authors have applied a new approach (an Environment-Wide Association Study -EWAS-) to data from Nurses' Health Study (NHS) to explore the effects of many foods and nutrients on Coronary Heart Disease (CHD) risk. The authors have included 62,811 women in the analysis (2,774 cases -4%- and 60,037 controls -96%-). With this technique, the authors identified only 16 food items and 37 nutrients that show statistically significant association with CHD risk. The paper is well-written and very comprehensive, and results and conclusions are very interesting, but the manuscript may be improved if the authors take into account the following:

1. Dietary patterns: Traditionally, nutrition research has focused on single nutrients or specific foods, as the current study, although individuals do not consume nutrients or foods in isolation. Several nutritional epidemiologic studies have shifted to dietary pattern analysis, which describes the overall diet (Am J Clin Nutr.2015;101:899{900). Overall diet capture food groups, nutrients and, more important,their combination and variety; Thus, dietary pattern may allow a closer relationship analysis with health outcomes. The authors should include a paragraph in the introduction on this issue, and

explain the usefulness of their food and nutrient analysis in front other approaches such as dietary pattern analyses.

We thank the Referee for the gracious comments. In the following, we address all the recommendations offered by the Reviewer. We hope that the revised manuscript will clarify any potential misunderstanding our previous formulation may have created.

1. We fully agree that the dietary pattern approach, as opposed to single-nutrient approach, should be mentioned in the manuscript. While dietary patterns are ideal as a means to develop nutritional guidelines [1], the strength of EWAS relies on agnostic discovery of new signals for further experimental or mechanistic validation. We modified the introduction accordingly and cited the work recommended by the reviewer (line 46 to 50).

2. Polyphenols: The authors have performed a deep analysis on the relationship between several nutrients and CHD outcomes, including several subclasses of polyphenols (Tables). However, the role of these compounds on CHD have not been included in the results and consequently in the discussion. This reviewer believes that polyphenols merit a greater attention (Biochem Pharmacol.2018;156:186-195).

2. As the Referee mentioned, we covered a wide range of polyphenols. Among 44 polyphenol-related exposures, 17 were found to be significantly associated with lower CHD risk at a 0.05 threshold of P-value. However, only isorhamnetin and apigenin remained significant after correcting for multiple testing ($HR < 0.92$; $P\text{-value} < 2 \times 10^{-4}$; lines 285 and 286). We fully agree with the Referee for the important role that polyphenols potentially have on health, hence following the Referee's recommendation, we added a section in the Supplementary Information (Section 6) to discuss our broader findings about polyphenols.

3. Potassium and Magnesium: Both ions have important effects on health and also merit to be considered.

3. We agree that potassium and magnesium have potential impacts on health; however, we did not find potassium to have a statistically significant association with CHD risk. We found magnesium

to have an inverse association with CHD risk (HR < 0.91; P-value < 4×10^{-3}); however, this association was no longer statistically significant after correcting for multiple testing.

4. Beer: In addition to white and red wines, beer also merits some attention in the manuscript (at least a sentence).

4. Regarding beer, we did not find a meaningful relation with CHD risk. As described in the discussion, there are many possible reasons as to why a true effect might be missed in this analysis (i.e., a false negative), including lack of power in the context of correction for multiple testing in the EWAS context or insufficient between-person variation. We did add a sentence discussing this issue, mentioning wine and beer (line 224 to 226).

We would like to thank the Referee again for the thorough reading of our manuscript. We found the recommendations very helpful, and trust that the Referee finds the revised manuscript appropriate for publication.

Comments by Reviewer #2

The manuscript by Milanlouei and Menichetti et al. explores how dietary factors associated with heart disease risks can be systematically analyzed using a framework inspired by Genome-Wide Association Studies. The authors dub this Environment-Wide Association Study (EWAS). The results are further analyzed by breaking down food items into nutrients. I think the paper is interesting, however, there are many unclear technical issues as well as issues in relation to disease definitions. It is positive that the authors validate their findings in the NHS-II cohort, it would strengthen the paper further if validation in the UK Biobank cohort was added, this cohort of 500k individuals includes both men and women. I [have] issues with the manuscript as is, which made it difficult to ascertain whether the associations are valid.

We thank the Referee for the valuable recommendations. We modified the main text to address each of these valuable suggestions. In the following, we offer a point-by-point summary of the changes.

Introduction

1. The introduction is well written, however, it could mention two of the larger prospective cohorts that also record diet (UK Biobank, Kadori Biobank). Only very little is mentioned about the underlying genetics of CVD.

1. We thank the Referee for bringing our attention to these two cohorts. Accordingly, we modified the introduction to discuss the UK Biobank and the China Kadoorie Biobank (line 88 to 91). We fully agree that a validation in these cohorts could strengthen our results; however, the dietary data in the UK and Kadoorie Biobanks are seriously limited: very few participants in the UK Biobank study provided more than two days of dietary recall, and the Kadoorie study has only a very small number of food frequency questions. We fully agree that genetic components are affecting CHD risk, a problem well-studied in the literature. Here, however, we focused on the environmental aspect of the disease and chose not to discuss the genetics of CHD, owing to word limitation and the primary purpose of this analysis.

Methods

The methods section appears to be incomplete in several ways: 2. The paper lacks a proper description of the two cohorts used. Number of participants, number of participants lost at each follow-up, average number of follow-up years etc. It was not until the Discussion that it was mentioned that the cohort consisted of women only.

2. We thank the Referee for pointing this out. Following the Referee's recommendation, we added this information to Section 1 in the SI (line 13 to 30). For reference, from 121,527 participants in NHS, we followed a standard procedure and excluded 47,949 from the analysis who dropped out from the study before 1986, who did not answer the dietary questionnaire in 1986, whose reported average energy intake was less than 600 or more than 3,500 kcal/day, who left more than 70 questions in the FFQ unanswered, or whose demographic data at baseline were missing. Moreover, 10,767 participants with a history of cardiovascular disease (CVD), diabetes mellitus, and cancer in the baseline year were excluded from the analysis. The average number of follow-up years for non-case and case individuals were 26 and 16 years, respectively. In NHS II, from 116,429 participants, 21,181 were excluded if they did not answer the dietary questionnaire in 1991, their reported average energy intake was less than 600 or more than 3,500 kcal/day, they left more than 70 questions in the FFQ unanswered, or whose demographic data at baseline were missing. In addition, 4,387 participants with a history of cardiovascular disease (CVD), diabetes mellitus, and cancer in the baseline year were excluded. The average number of follow-up years for non-case and case individuals were 22 and 14 years, respectively.

3. Ascertainment of CHD (p. 18): The definition of cases (acute myocardial infarction (MI) and fatal CHD) and non-cases: Is it understood correctly that cases survive the acute phase? Also, the annotation does not seem to be fully aligned as you write nonfatal MI in the Results section (p. 4).

3. We apologize for the confusion. As mentioned in line 434 to 436, a participant is considered a case if, at any point within the follow-up period, she reported development of nonfatal myocardial infarction or whose fatal CHD was documented by death records.

4. It is unfortunate that the all details of the underlying code are not made available (see below).

4. We apologize for not incorporating all code details. We modified our GitHub repository according to the Referee's suggestions.

5. Individuals with >70 patients unanswered questions removed. This is unfortunate and is likely to bias the analysis unless data is missing completely at random. See e.g. <https://www.ncbi.nlm.nih.gov/pmc/articles/PMC1839993/>. With the Cox regression model, it should be simple to use e.g. multiple imputation as implemented in mice to obtain an unbiased estimate. Furthermore, is this to be understood as you also removed individuals with incomplete follow-up? This would also be problematic, and bias the analysis a lot.

5. The FFQ used in NHS contains approximately 145 food items. The rationale behind excluding participants who left more than 70 items blank is that the obtained incomplete FFQ does not properly represent that individual's diet. The same criterion has been applied by previous studies [2–4]. It is correct that food items are not missing at random, and in a detailed follow-up of a sample, it was clear that the most common reason for leaving an item blank was that the person did not consume that food. Thus, blank items are considered zero when calculating nutrients. Only a small percentage left more than 70 items blank; since most of these involved sections missed or other serious errors, they were considered of questionable validity and excluded. In other cases of missing values in the dietary data, we used the *last observation carried forward* (LOCF) approach, a common method applied when working with time series data. Following the Referee's recommendation, we added related references to the manuscript for greater clarity (line 409).

6. Why censor individuals developing diabetes/cancer? Why not just include variables for diabetes/cancer in the model?

6. For participants who developed diabetes mellitus or cancer during the follow-up, we did not censor them, but we did not further update their dietary record because these conditions can induce major changes in diet and lifestyle habits after diagnosis. As we mentioned in lines 415 and 416, we

utilized this approach in order to minimize the effect of reverse causation, a common approach in nutritional epidemiology [5-8].

7. In Supplementary Figure 2, this is indicated to be a case-control study. This is not a case-control study, it is a cohort study. There are no controls, only individuals not developing the outcome.

7. We thank the Referee for pointing this out and we modified Figure 2 in the Supplementary Information accordingly.

8. Population (p. 17 + 18 and Figure SI. 1). Handling of cases and noncases. It reads as if cases must develop acute myocardial infarction or fatal CHD as their first disease. Otherwise the NHS participant is classified as a non-case. What is the underlying rationale here? Is it primarily a statistical motivation or a phenotypic? In the latter case, the workflow is a bit misleading unless you also present the counts on how many NHS participants develop diabetes and the acute myocardial infarction, for instance. A priori one would think they ought to be taken into account.

8. It is correct that a participant is a non-case until she develops acute MI or fatal CHD as this is the definition of case for this analysis, at which point the follow-up is censored. Women who develop diabetes are documented and this status is indicated as a covariate but follow-up continues. Following the Referee's recommendation, we expanded Section 1 in the SI, indicating the number of participants for whom we stopped updating the dietary records.

9. Please include a description of how many women were removed at each step.

9. We fully agree and we modified Section 1 in the SI to reflect this description.

10. "our own analyses of fatty acid composition" - Please describe this. Furthermore, including the compiled list of nutrients by food item as a supplementary file would be very useful for future studies.

10. Fatty acid and other nutrient values were calculated based on the Harvard University Food Composition Database, which is updated regularly using external publications and direct analysis of fatty acids in commonly used margarines and processed foods to take into account changes in manufacturing processes. We revised the manuscript accordingly to make clear that fatty acid composition analysis is not part of this study (line 429 to 430). A compiled list of nutrients by food item is available online and following the Referee's suggestion, we added the relevant reference to the manuscript (line 423).

11. "Ascertainment of CHD" - can you please confirm if this was done as part of the current study, or just in general for the Nurses Health Study?

11. This is a general procedure used in the Nurses' Health Study. This procedure has been performed as an ongoing part of the Nurses' Health Study and has been the basis of many previous published analyses.

I also have some concerns with the statistical analysis:

12. When using the extended Cox model, did you include a baseline risk for each participant to consider having a different number of data points for each individual?

12. A baseline risk dependent on the number of data points per individual is a modeling choice usually considered when participants may drop out of the study for several reasons not captured by the model, introducing potential selection bias. The number of data points per individual is then consistent with a hidden variable, taking into account all the potential effects influencing the observability of certain groups. However, as Figure 1 demonstrates, in NHS the number of data points per individual is a strong predictor of CHD, meaning that a limited permanence period in the study is mainly associated with the onset of CHD, with no real need for additional explanatory variables.

13. Please indicate the underlying time for the Cox model.

13. The underlying time for the Cox model is the time on study. The time scale for the analysis was measured in months since the start of the questionnaire cycles which is equivalent to age in months. We modified the Methods section accordingly to reflect the underlying time for the Cox

Figure 1: Number of data points per individuals, stratified as cases and non-cases

model (lines 447 and 448).

14. Age was modeled as a linear covariate. What is the rationale behind this? Why not include e.g. a spline to take into account non-linear patterns in the association between CVD and age?

14. We thank the Reviewer for raising this question. Modeling age as a linear covariate is a common statistical choice. However, we also tried a spline of degree four and the outcomes were almost identical to the linear model (data not shown).

15. Not being fully familiar with NHS, have hormone contraception been recorded at other time points? This could also be very relevant to adjust for.

15. We have data on both hormonal contraception (birth control pills) and hormone therapy after menopause. The studies done using NHS data generally have not controlled for contraceptive use because few women used them during the follow-up period. We adjusted our analysis for hormone therapy.

16. When using the cumulative average of food intakes, did the authors attempt to do any outlier detection? Or are these results presented somewhere?

16. In the FFQs, for each dietary questions, nine responses were possible, from “almost never” to “six or more times per day”. We converted the frequency responses to the number of servings per day for each food item. This structure inherently prevents having outliers for a single dietary item. In the bigger picture, we avoided outliers by excluding participants whose reported average energy intake was less than 600 or more than 3,500 kcal/day (this was approximately 5% of women).

17. "We selected the set of adjusting variables based on their contribution in explaining the variability in CHD risk" - What exactly does this mean? Is this analysis part of this work?

17. We apologize for the lack of information. We were provided with a set of potential confounders and adjusting variables by the cohort PI's. The suggested list was based on a huge body of knowledge accumulated in previous studies from the Nurses' Health Study. We excluded those which did not have any effect on the analysis owing to their having a very low variability across the cohort and not providing informative power (lines 454 and 455 and line 337 to 340).

18. How was the proportional hazard assumption verified?

18. We checked the correlation between the Schoenfeld residuals and survival time, where a correlation close to zero supports the proportional hazards (PH) assumption [9]. We reported the p-value associated with the PH assumption test in Table 1 in the manuscript.

19. The authors cite a study used to correct for confounders in case-control studies. However, this is not a case-control study. It is a cohort study. Case-control studies are typically enriched for cases, compared to cohort studies. Can the authors please justify how a case-control procedure generalises to a cohort study, and if it introduces any bias?

19. The idea provided by [10] is that whenever a permutation procedure is used to make inference about the significance of a test, one should keep in mind that the effect of confounders should be kept in place. For example, when we test the association between fat intake and CHD risk, we cannot simply shuffle the CHD status among participants and recalculate the association between

fat consumption and CHD risk. Rather, we want to take into account that regardless of the amount of fat that an individual consumes, an older individual who is a smoker and does not exercise should have a higher chance of developing CHD, compared with a young individual who does not smoke and does exercise regularly. This concept itself is independent of study design and we, therefore, applied it to our analysis.

20. Also, 1,000 permutations seem very little considering the power of modern computers.

20. Regarding the number of permutations, we looked into the convergence to the number of significant tests after each iteration. After ~ 300 permutations, the number of significant tests did not change; however, we continued the permutations up to 1000 to have accurate estimations for FDR (Figure 2). Using 30 cores and 200G RAM, it takes approximately 30 minutes to run one iteration of permutations.

Figure 2: Number of significant tests as a function of the number of permutations

21. Instead of assuming each test is independent, which is it not, could the authors do e.g. a PCA decomposition of the individual x food matrix to approximate how many effective tests the authors are performing? See e.g. <https://journals.plos.org/plosgenetics/article?id=10.1371/journal.pgen.1006711>. Also, please note that the Bonferroni correction is not used in a GWAS setting

as described. Instead, the number of individual tests, due to correlations in the genome, have been estimated.

21. In our framework, we do not assume that tests are independent; rather, we use a procedure similar to [11], where the correlated structure of the tests was taken into account [12,13]. Westfall and Young [14] discussed how permutation-based methods exploit the correlation structure between tests in detail. We acknowledge the existence of other methods such as principal component analysis to find the number of effective tests, followed by a Bonferroni-like *Dunn-Šidák* correction [15]. Sometimes these methodologies are preferred over permutation procedures which are more computationally intensive; however, they are also extremely conservative, being modifications of the Bonferroni correction, controlling for the family-wise error rate (FWER) [15,16]. Considering the small effect sizes characterizing nutritional epidemiology, we opted for controlling the false discovery rate (FDR), in line with previous EWAS studies [11,17–20].

We wish to thank the Referee for bringing this issue to our attention. We, therefore, modified the manuscript as to the application of Bonferroni and *Dunn-Šidák* corrections in a GWAS setting (SI, Section 4, line 41 to 43). Both the methodologies select 31 significant tests.

22. Ascertainment of Diet. "Nine responses were possible, ranging from almost never" to six or more times per day." We converted the frequency responses to the number of servings per day for each food item." How did you transform the categorical variables to continuous variables? While the frequencies between 1-5 times/day are simple, the conversion for "almost never", is unclear, as 0? And six or more? Was it transformed to 6? "We calculated daily intake of nutrients". How did you decide on the units for nutrient amounts? For that you need to specifically know the food intake amount.

22. The FFQ collects the frequency of consumption of one serving size for each food item. These values are converted to the number of serving sizes consumed per day. Never, or less than once per month, 1–3 per week, 1 per week, 2–4 per week, 5–6 per week, 1 per day, 2-3 per day, 4-5 per day, and 6+ per day are converted to zero, 0.07, 0.14, 0.43, 0.79, 1, 2.5, 4.5, and 6, respectively. Serving sizes for food items are converted to weight internally. Using the food composition database, the weight of consumed food is translated to the weight of each nutrient. A detailed description of

the food-frequency questionnaire and documentation of its reproducibility and validity have been published elsewhere [21–23]. Prompted by the Referee’s question, we added related references to the manuscript (lines 419 and 420).

23. The nutrient content of each food depends on many factors, of which the brand is important. In the cited paper’s (reference 83) database one can search for a specific brand of food. How did you manage to obtain the nutrient information of unspecific foods, such as: apple juice, hot dog, etc. . . without searching for specific brands? It is well known that different brands use a broad range of different food additives which are not part of the nutrient content but have the potential to induce adverse reactions in the organism, which in turn could be associated to development of CHD. Some examples are the food additives nitrates and nitrites which have been associated to carcinogenicity, and thyroid hormone disruption (Bouvard V, Loomis D, Guyton KZ, et al.; International Agency for Research on Cancer Monograph Working Group. Carcinogenicity of consumption of red and processed meat. *Lancet Oncol.* 2015;16(16):1599{1600 Tonacchera M, Pinchera A, Dimida A, et al. Relative potencies and additivity of perchlorate, thiocyanate, nitrate, and iodide on the inhibition of radioactive iodide uptake by the human sodium iodide symporter. *Thyroid.* 2004;14(12):1012{1019) The food additives should also be added to the study, in order to fully cover the spectrum of exposure. Besides, these food additives could be the ones behind the underlying adverse effects. Furthermore, there is several indirect food additives, which are chemicals that are not directly added to the food but are put in contact with it among others through the packaging process. These include bisphenol, phthalates, etc. Although they cannot be taken into account fully, since they are widely unknown, it is important to mention them as a limitation of these type of study.

23. We did have data on brand for some commonly consumed foods including breakfast cereals and margarines and took them into account in our calculations. We fully agree with the reviewer that food additives and other chemicals added to foods during the packaging need to be taken into

account to have a broader picture of the effect of diet on health. However, these chemicals are not always available in the food composition databases. We added to the Conclusions as a limitation that details of other additives and contaminants could not be assessed and cited the works proposed by the Referee (line 329 to 333).

24. "our own analyses of fatty acid composition". What does this mean? Where do you describe this analysis?

24. As we mentioned earlier in question 10 here, fatty acid and other nutrient values were calculated based on the Harvard University Food Composition Database, which is updated regularly using external publications and direct analysis of fatty acids in commonly used margarines and processed foods to take into account changes in manufacturing processes. We revised the manuscript accordingly to make it clear that fatty acid composition analysis is not part of this study (lines 429 and 430).

25. The authors fail as far as I can see to explain the time-dose use. The authors used a cumulative average food over a 2 year interval. This approximation fails to handle dose of exposure changes over time, which often are critical if one wants to understand the effects of the nutrients in disease development. The intake of a nutrient does not necessarily produce an adverse outcome, unless there is a continued specific dose exposure. For example, alcohol has been associated in this study with a protective effect. However, in high doses, alcohol has previously been associated with disease development (Alcohol consumption and risk of stroke: a meta-analysis, Reynolds K, Lewis B, Nolen JD, Kinney GL, Sathya B, He JAMA. 2003 Feb 5; 289(5):579-88.) .

25. We fully agree with the Referee on the importance of the time-dose use. Please note that at each time step during the follow-up, we calculated the average of all previous dietary data points up to a specific time so that the last observation is the average over the entire follow-up period. The reason behind utilizing this approach is that CHD, is a longitudinal disease in the sense that the development or prevention of CHD occurs as a results of years of following a specific lifestyle. In contrast, a disease like type 2 Diabetes is much more dependent on the short-term diet (although not exclusively so). The cumulative average model takes advantage of all prior data and thus should provide a statistically more powerful test of an association of cumulative

exposure. Moreover, the average value of repeated measures approaches the true value; that is, the law of large numbers applies, and therefore, the random measurement error will be reduced [21]. Regarding the effect of alcohol consumption, within the range of alcohol intake consumed by this population, we observed an inverse association between alcohol consumption and CHD risk. It is worth noting that the overall level of alcohol consumption is not very high in this population study.

26. There seems to be several issues with the code, including the lack of a data file that could be used to perform a test run. The authors also fail to define the format of the input data as far as I could see. The rest of the code seems well formulated and commented apart from these aspects: 1) The authors fail to explain what the first function does, "addToBase"; 2) Re "na.locf", this step adds the previous value to all the following rows which contain NA. Without a data table one cannot assess whether this is a biased manual imputation, or an imputation that correlates to what the authors have defined in the Methods section of the paper; 3) "tempvar <- c(covar, "calorconv)". What does "calorconv" refer to?

26. We do apologize, again, for the incompleteness of the code which has now been corrected. We added a data file to perform a test run. We also clearly defined the format of the input data. 1) We explained the functionality of "addToBase". 2) This line will be removed. 3) We corrected this line. Please refer to github.com/soodimilanlouei/EWAS-NHS to see the modifications.

Results

27. Figure 1: The knowledge graph (p. 5). The difference between the diamonds seems somewhat unclear. E.g. What is the difference between coronary heart disease (CHD) and coronary artery disease? And what phenotype does cardiovascular disease (CVD) include that is not included in any other diamond. According to the Supplementary Information section 1 there were 124 negative and 45 positive associations documented in relation to heart-related complications. However, translation to the knowledge graph seems fuzzy.

27. In constructing the knowledge graph, we report the disease phenotypes as mentioned in the literature, without a further disambiguation step. In the context of NHS, CHD refers to non-fatal

MI and fatal coronary heart disease; coronary artery disease (CAD) also refers to non-fatal MI and fatal coronary artery disease. Cardiovascular disease (CVD) is defined as a composite of coronary artery disease and nonfatal or fatal stroke. In response to the Referee's question, we modified the figure's caption to include this information. A negative association means that a higher level of an exposure was associated with a lower risk for a disease; a positive association means the opposite. Negative associations are shown with green links and positive associations are represented with red links.

28. Figure 2a: I do not understand this figure. Please provide a clearer explanation in the Statistical Analysis section instead. Again, Cox estimates hazard, not odds.

28. We apologize for the confusion. We modified Figure 2-a and the Statistical Analysis section accordingly. We thank the Referee for bringing our attention to this incompleteness.

29. "The minimum statistical power for detecting the smallest absolute effect size was 0.59, which is considered to be a moderate to high level of power in clinical studies" - post-hoc power calculations, or "observed power" are not appropriate.

29. Previous studies [19, 24] have used post-hoc power calculations but we fully agree with the Referee that post-hoc power analysis is not appropriate here. We have, therefore, removed this section from the manuscript.

30. "Traditional epidemiological studies are limited to the detection of a single exposure (food or nutrient) in relation to CHD". That is absolutely not true. See e.g. <https://academic.oup.com/ije/article/36/3/600/652363>.

30. We modified the manuscript accordingly and added the recommended work to the manuscript (line 46 to 50).

31. Please move the description of the network construction to the Methods section.

31. We moved the description of the network construction to the Methods section (line 383 to 403).

32. "Below, we discuss both non-validated and validated associations found by EWAS in the context of previous literature.". There is nothing below.

32. We thank the Referee for pointing this out. We modified the manuscript accordingly.

Conclusion

33. "Third, while we included only the adjusting variables with statistically significant effects on CHD risk rather than selecting them subjectively, our study remains limited to the common covariates usually considered when exploring diet-CHD associations". Again, what selection was made exactly?

33. As we discussed in question 17, we were provided with a set of potential confounders and adjusting variables by the cohort PI's. The suggested list was based on a body of knowledge accumulated in previous studies from the Nurses' Health Study. We excluded those which did not have any effect on the analysis owing to their having a very low variability across the cohort and not providing informative power.

34. "Indeed, our use of a large longitudinal dataset with a long follow-up period and a sufficient number of subjects helped us achieve sufficient statistical power to detect even relatively small effect sizes". Is this necessarily a strength? Very small effect sizes often have little or no real life value. It would be good to discuss how exactly the hazard ratios should be interpreted. E.g., if I drink one unit of alcohol, how much does that affect my risk? What about a carrot?

34. Given that CHD is the most common cause of death in the US and worldwide, even very small effect sizes can be very important, especially if they can be modified modestly and at minimal cost. The value of an effect also depends on other effects on other outcomes, balancing all of which is beyond the scope of our analysis. However, the EWAS approach is, in fact, a screening methodology that ranks the effect sizes and can be used to select those exposures with most robust signal,

discarding others that would (only) pass a single-test analysis.

In line 144 to 148, we discuss how to interpret the hazard ratios for each exposure.

35. Conclusion and Future Work (p. 15 + 16). It is surprising that smoking is not included in the discussion. Is it considered part of "air quality" or is it left out for some seemingly less obvious reason?

35. Smoking is of one the confounders for which we adjusted our analysis. Smoking is represented as a categorical variable with five levels (never; past smoker: ≤ 5 pack-years, 6-20 pack-years, or ≥ 21 pack-years; current smoker: ≤ 5 pack-years, 6–20 pack-years, or ≥ 21 pack-years). However, understanding the effect of smoking on CHD risk was not the main goal of this analysis, and thus, it has not been discussed in the manuscript.

36. The authors have only focused on CHD and have eliminated any other comorbidity from the analysis, which can be correlated to the development of this disease. The authors should perform a competing risk analysis in which there are several possible outcome events, in this case any of the diseases shown in Figure 1.

36. We fully agree with the Referee that multiple comorbidities do exist. In this specific study, we aimed to understand the effect of diet solely on CHD risk. This is also the reason behind the fact that whenever a participant developed other diseases, such as diabetes mellitus or cancer, she would be classified as a noncase, with no further update of her dietary records but continued follow-up. Our work is the very first attempt to apply an EWAS approach to study CHD in the NHS data; consequently a competing risk analysis would have been harder to compare with previous single-nutrient results in the literature. We acknowledge the relevance of a competing risk analysis; however, this would require a whole new pipeline to be developed and tested, and will be the subject of further study.

37. Supplementary: line 195–198. I do not understand how the approach utilises the structure in dietary information. As far as I can read, only the

confounding variables are used.

37. We apologize for the incompleteness. We added references to the SI (line 59) where it has been discussed in detail as to how the permutation procedure accounts for multiple testing under dependence. The use of confounding variables is one step before applying the permutation procedure. We used the confounding variables to estimate the probability of developing CHD regardless of the dietary intake, and later, permute the CHD status based on these obtained probabilities.

Minor

38. spelling error: "hazard ratio for one stabdard deviation" "We used Cox regression to estimate the odds of developin" - The Cox regression model estimates hazard ratios, not odds.

38. We thank the Referee for pointing out the spelling errors. We modified the manuscript accordingly.

Lastly, we would like to thank the Referee for the thorough and detailed reading of our manuscript. We found the Referee's recommendations extremely constructive and we hope that the revised manuscript has addressed most of the Referee's legitimate observations.

Comments by Reviewer #3

This is an important and timely paper that aims to overcome the limitations of most literature on nutrition and health, in particular the single-item approach. They apply advanced statistical methods to disentangle the effects of nutrients and foods taking their inter-correlations into account. They also validate the results in a separate dataset. As such, this is a nice and well-written paper, though the authors are aware it is only a first step in advancing the field: - We know a limited number of components in food (the tip of the iceberg). I like the analogy with dark matter - There is some degree of mismeasurement that implies misclassification - Foods act via nutrients and other components, meaning that rather than a correlational analysis next steps could include mediation analysis (as they rightly point out, there is a many-to-many relationship between foods and their constituents) These advancements could be left for next papers.

Main comments:

1. I would have expected a pattern analysis, and in particular a visual representation of inter-relationships like in an "exposome globe"

We thank the Referee for the excellent summary of the paper's contribution to the field. We modified the main text to address each of the valuable suggestions.

1. We agree with the Referee that a pattern analysis would add value to our study. However, owing to the word limit in the main manuscript, we could only add a section in the Supplementary Information (Section 6) dedicated to correlation analysis using a correlation matrix. To have an alternative visual representation of inter-relationships, we developed a correlation globe, similar to [25], but due to the dense nature of the inter-correlations and the high number of exposures, the resulting exposome globe was visually challenging to interpret; therefore, we chose to show these relations in a correlation matrix where we observe the natural clustering more easily.

2. The example of Kolonel and papaya looks a little out of place, being only one example that might be due to chance.

2. We acknowledge that this might be due to chance (other studies have not had enough papaya consumption to evaluate this relationship; the Kolonel study was performed in Hawaii). However, the point is still germane even if the original finding was due to chance.

3. Notice that in the paper by Merritt coffee was inversely associated with endometrial cancer.

3. We thank the Referee for pointing this out. We added the direction of the association found by Merritt et al. to the manuscript (line 63).

4. In the introduction the main limitation of studies seems to be related to sample size, but there are many others: misclassification of food habits, intra-individual variation, temporal variation, lack of repeat measurements, etc

4. We fully agree, and hence, included these limitations to the introduction (line 68 to 71).

5. the limited statistical power of previous studies is not absolute. EPIC and UK Biobank are large cohorts.

5. We fully agree with the Referee that the statistical power of previous studies is not absolute. While EPIC and UK are large studies, regarding dietary data collection, they did not collect these data longitudinally, practically making them a cross-sectional study when it comes to studying diet, and the dietary data in those studies are limited comparing with the NHS in terms of the follow-up period and the number of participants whose dietary data have been collected. CHD itself is a longitudinal disease in the sense that the development or prevention of CHD occurs as a result of years of following a specific lifestyle. In contrast, a disease like type 2 Diabetes is much more dependent on the short-term diet (but not exclusively so). Generally, longitudinal designs are at times favored over cross-sectional studies because they allow the detection of within subject variation over time and typically have higher statistical power [26–29]. In other words, since there are repeated observations for each individual, longitudinal studies are highly relevant and have more power than cross-sectional observational studies, as they are able to exclude time-invariant

unobserved individual differences and to observe the temporal order of events [30].

6. I am not completely persuaded by the analogy with GWAS. In GWAS one is interested in single gene variants, confounding is limited and the structure of the data is completely different from EWAS, being dictated by haplotypes.

6. We fully agree with the Referee that there are structural differences between GWAS and EWAS. Yet, methodologically, EWAS takes inspiration from GWAS, which have been used to investigate the relationship between genome-wide variability and a disease [11]. Both GWAS and EWAS share the similar goal of detecting signals agnostically in a correlated high dimensional space. As in GWAS, multiple testing was controlled for and significant associations were validated in another cohort. GWAS and EWAS have enabled the generation of new hypotheses regarding the relationship of genetics and environment to a disease of interest. We acknowledge that there are many limitations to EWAS and in response to the Referee's observation, we have described them extensively in the discussion.

We would like to, again, thank the Referee for evaluating our paper and providing constructive insights that have improved the final manuscript.

Comments by Reviewer #4

1. This paper is using an EWAS approach to identify effects of diet on coronary heart disease in women (CHD) using the Nurses Health Study. Observational studies of the effects of diet on health are well-known to be difficult to interpret because of bias from confounding, i.e., it is difficult to separate the attributes of the individual from the effect of the exposure. To put it another way many, many factors may determine both diet and health, i.e., are common causes of diet and health, and these factors will bias the observed associations of diet with health unless they are fully accounted for in the analysis. Prime factors that might affect diet and health include all aspects of socio-economic position, including parental attributes, household income, wealth, familial status and level of education, which are impossible to measure comprehensively and precisely. To give unbiased estimates of the effects of diet on health the analysis must be fully adjusted for all factors that could bias the observed associations, i.e., all confounders of the association of diet on CHD, where confounders are defined as common causes of exposure and outcome.[1] Given confounding is a causal concept,[1] confounders cannot be identified reliably from observational data.

Each association in this study should be adjusted for all factors that could confound the association of diet with health, i.e., common causes of diet and CHD. In this study the concept of confounding is not used, and instead the analyses are adjusted for covariates (figure 2) identified from the data (methods). The covariates included in the analysis do not include key confounders, such as all aspects of socio-economic position, but do include factors that are unlikely to be confounders, such as aspirin use, high blood pressure and elevated cholesterol. As such, each of the associations in the EWAS are highly likely to be biased by unmeasured confounding and potentially by mediators. The EWAS analysis of these associations is thorough and comprehensive, but it is not designed to be nor is capable of rectifying the fact that each association included is potentially biased, meaning that the findings cannot be interpreted.

We thank the Referee for the valuable recommendations. We modified the main text to address each of the suggestions. In the following, we offer a point-by-point summary of the changes.

1. We agree that the notation of adjusting for covariates might be misleading and the concept of confounding should be used instead. In response to the Referee’s recommendation, we changed the manuscript accordingly, showing that we are, indeed, adjusting our analysis based on their potential confounding effects (lines 454 and 455).

We included aspirin in the model as a potential confounding variable because it is associated with lower risk of CHD in previous studies [31–34] (although recent data challenge its broad use in primary prevention) and, as a health-conscious behavior, could also be associated with dietary variables. The roles of high blood pressure and high cholesterol are potentially complex because these may, in part, mediate the effects of diet, but they can also be confounders because the diagnosis can alter diets. Moreover, they are known to be associated with higher risk of CHD [35,36]. Including them in the model, as we did, probably underestimates the effects of diet.

Regarding socio-economic status (SES), our study population comprises registered nurses with a relatively high homogeneity in educational attainment and socioeconomic status. In the manuscript, we acknowledge this homogeneity as a possible limitation of our study (see Conclusions), limiting the generalizability of our findings to more socio-economically diverse populations.

Nevertheless, prompted by the Referee’s observation, we investigated the role of SES by including census tract family median income in our analysis [37,38], and added Section 7 in the supplementary material to summarize our findings. The results are consistent with our previous analysis, confirming 43 out of 53 significant exposures originally found without SES, with only the marginal associations being affected by the additional correction. The ranking of the p-values is highly consistent, with a Spearman correlation coefficient equal to 0.9733.

Notably, although census tract family median income is a strong predictor of CHD in many studies as it incorporates many difficult to measure factors, it was only weakly associated with risk of CHD in our study population after accounting for diet, smoking, and the other behavioral factors that we included in our model. This does not mean that SES is not important,

but, rather, that the effects of SES on risk of CHD are largely mediated by diet and lifestyle factors in this population. The finding that some marginally significant associations became non significant after controlling for SES is very plausibly due to “over control” for a variable that itself is not a direct cause of disease, such as SES, but rather a determinant of diet and other behaviors.

Although our data suggest minimal residual confounding by SES variables, causality is always complex to document in situations where randomized trials are difficult or impossible to conduct, as for most of the relationships that we evaluated. This does not mean that we are obligated to live in ignorance, a long standing topic beyond the scope of our paper. We are clear that this is an exploratory, hypothesis-generating analysis; the identification of many well-known relationships is an important proof-of-principle for this approach. Associations that have been identified need to be evaluated using additional approaches appropriate for the specific relationship.

Specific comments

Abstract

2. Please give a more nuanced interpretation of the results of dietary epidemiology. Please avoid the use of the word effect when describing observed associations. Please clarify the extent to which this study has replicated the effects of diet on health established from a systematic review of experimental or quasi experimental studies, such as randomized controlled trials or instrumental variable analyses.

2. In the manuscript, we mentioned that we used the term *beneficial* (or *negative*) when a higher level of an exposure is associated with a lower CHD risk. Similarly, we used the term *harmful* (or *positive*) when a higher level of an exposure is associated with a higher CHD risk. We made this choice for simplicity and it should not be confused with a causal relationship. However, we changed this narration to eliminate any further confusion, following the Referee’s recommendation. We pursued a type of validation consistent with those commonly applied by the EWAS community [11, 17–19, 24, 39]. Given the extensive number of exposures considered in these studies, the replication of their findings is limited to other data collections with a compatible panel of exposures. Unfortunately, this dramatically reduces the possible validation data sets, forcing the authors frequently to work on different data collections within the same program (e.g. NHANES,

NHS). Despite the inherent methodological bias, these validations capture those associations that are reproducible in independent samples of the population. We added a paragraph to the manuscript in which we clarified the limitations of the present validation.

Regarding a possible validation with an orthogonal study design, we expect this would require a time-consuming matching of the panel of exposures and a careful investigation of the compatibility of the cohorts, in order to avoid misleading results as documented in the case of several meta-studies [40].

3. Please make it clear that the integrity of the study depends on assuming no unmeasured confounding of the associations of dietary items with health, and list in the abstract the confounders considered so that the reader can make their own assessment of the validity of the findings.

3. We agree, hence, in the Conclusions, we discuss the limitations of this analysis in detail. In particular, we note that our study remains limited to the common confounding variables usually considered when exploring diet-CHD associations. Hence, residual confounding is still possible as a result of ignoring the adjustment for other important risk factors (line 18 to 20 and line 337 to 341).

Introduction

4. Please give a more nuanced assessment of the achievements of dietary epidemiology. Many vitamins and minerals identified from dietary epidemiology as protective were not validated in randomized controlled trials, and some even turned out to be harmful [2].

4. We agree with the Referee for the need for this assessment. We have hence noted this point in the Introduction (line 78-81).

5. Please explain more clearly why EWAS overcomes problems of correlated exposures and unmeasured confounding.

5. We apologize for the confusion. EWAS does not overcome the problems of unmeasured confounding and there is no indication of such a claim in the manuscript. EWAS, similar to other statistical procedures, is susceptible to bias due to unmeasured confounding. We have

modified this section for clarity and thank the Referee for pointing this out (line 337 to 346). To account for multiple testing, we utilized a procedure similar to [11, 17, 20] where the correlated structure of the tests was taken into account [12, 13]. Westfall and Young [14] and Efron [12] discuss in detail how permutation-based methods exploit the correlation structure between tests. In a nutshell, permuting the CHD status and the time of developing CHD among participant destroys any true differences between non-case and case participants, thereby enforcing the null hypothesis. Since rows are permuted intact, the correlation structure between the columns (i.e., the exposures) is maintained. Next, to estimate the number of false discoveries, we created a null distribution of Cox model P-values by randomly shuffling the CHD status 1,000 times and recomputing the P-values. Accordingly, we estimated the FDR to be the ratio of the proportion of results that were called significant at a given level α in the null distribution and the proportion of results called significant from real tests. In the FDR calculation, we found the rate that significant features are truly null empirically, without relying on the assumption of independence among tests. We added related references to Section 4 in the SI for greater clarity (line 59).

6. Please give evidence for the statement ".we consistently recover prior knowledge about diet-disease relationships", and clarify that this means recovering the results of randomized controlled trials, rather than replicating the previous observations from the Nurses Health Study.

6. As discussed earlier, we pursued a type of validation consistent with those commonly applied by the EWAS community [11, 17–19, 24, 39]. Given the extensive number of exposures considered in these studies, the replication of their findings is limited to other data collections with a compatible panel of exposures. Unfortunately, this dramatically reduces the possible validation data sets, forcing the authors frequently to work on different data collections within the same program (e.g. NHANES, NHS). Despite the inherent methodological bias, these validations capture those associations that are reproducible in independent samples of the population. In response to the Referee's recommendation, we added a paragraph to the manuscript in which we clarified the limitations of the present validation. Regarding a possible validation with an orthogonal study design, we expect this would require a time-consuming matching of the panel of exposures and a careful investigation of the compatibility of the cohorts, in order to avoid misleading results as documented in the case of several meta-studies [40].

Methods

7. Please use the up-to-date definition of a confounder

7. We took the Referee's recommendation and changed the manuscript accordingly (lines 454 and 455).

Results

8. Please amend the box in Figure 2a from "adjusting covariates" to "adjusting confounders" and amend the analysis accordingly throughout.

8. We modified both the figure and manuscript, accordingly.

Comparison with the previous literature

9. This section should give a comparison with evidence from randomized controlled trials and quasi-experimental studies. There is little value in showing that EWAS replicates findings known to be uninterpretable due to bias. Validation needs to be using data and designs which are not open to the same biases as this study, so observational analysis of Nurses Health Study II cannot provide validation.

9. As we mentioned above, we pursued a type of validation consistent with those commonly applied by the EWAS community [11, 17–19, 24, 39]. Given the extensive number of exposures considered in these studies, the replication of their findings is limited to other data collections with a compatible panel of exposures. Unfortunately, this dramatically reduces the possible validation data sets, forcing the authors frequently to work on different data collections within the same program (e.g. NHANES, NHS). Despite the inherent methodological bias, these validations capture those associations that are reproducible in independent samples of the population. We added a paragraph to the manuscript in which we clarified the limitations of the present validation.

Regarding a possible validation with an orthogonal study design, we expect this would require a time-consuming matching of the panel of exposures and a careful investigation of the compatibility of the cohorts, in order to avoid misleading results as documented in the case of several meta-studies [40].

Conclusion and Further Work

10. Please explain that this study is open to unmeasured confounding using an up-to-date definition of confounding with a more realistic assessment of the effects of unmeasured confounding.

10. We agree and we addressed this concern in previous sections (questions 1,3, and 5).

11. Please consider whether the other major source of bias in observational studies, i.e., selection bias

11. We agree, hence, we discuss a number of limitations of this study in the Conclusion section. Specifically, selection bias does not apply to our study. Selection bias occurs when there is a systematic difference between either those who enrolls in the study and those who do not, or those in the treatment group of a study and those in the control group. The former affects the generalizability and the latter influences the comparability between groups. The Nurses' Health Study is a prospective study that was designed to minimize susceptibility to these biases.

12. In view of these limitations please do not interpret this study as a guide to the causal effect of diet, but instead contextualize it within the more recent evidence from randomized controlled trials and mendelian randomization studies.

1. Bareinboim E, Pearl J. Causal inference and the data-fusion problem. *Proceedings of the National Academy of Sciences of the United States of America*. 2016;113(27):7345-52. PubMed PMID: 27382148.

2. Lawlor DA, Davey Smith G, Kundu D, Bruckdorfer KR, Ebrahim S. Those confounded vitamins: what can we learn from the differences between observational versus randomised trial evidence? *Lancet (London, England)*. 2004;363(9422):1724-7. Epub 2004/05/26. doi: 10.1016/s0140-6736(04)16260-0. PubMed PMID: 15158637.

12. We addressed this concern in the Conclusions and added the recommended references (line 341 to 346).

We would like to thank the Referee for the insightful suggestions which helped us improve the manuscript.

References

- [1] Tapsell, L. C., Neale, E. P., Satija, A. & Hu, F. B. Foods, nutrients, and dietary patterns: interconnections and implications for dietary guidelines. *Advances in Nutrition* **7**, 445–454 (2016).
- [2] Gates, M. A. *et al.* A prospective study of dietary flavonoid intake and incidence of epithelial ovarian cancer. *International Journal of Cancer* **121**, 2225–2232 (2007).
- [3] Conen, D. *et al.* Caffeine consumption and incident atrial fibrillation in women. *The American Journal of Clinical Nutrition* **92**, 509–514 (2010).
- [4] Guasch-Ferré, M. *et al.* Nut consumption and risk of cardiovascular disease. *Journal of the American College of Cardiology* **70**, 2519–2532 (2017).
- [5] Stampfer, M. J., Hu, F. B., Manson, J. E., Rimm, E. B. & Willett, W. C. Primary prevention of coronary heart disease in women through diet and lifestyle. *New England Journal of Medicine* **343**, 16–22 (2000).
- [6] Hu, F. B. *et al.* Frequent nut consumption and risk of coronary heart disease in women: prospective cohort study. *BMJ* **317**, 1341–1345 (1998).
- [7] Rimm, E. B. *et al.* Folate and vitamin B6 from diet and supplements in relation to risk of coronary heart disease among women. *JAMA* **279**, 359–364 (1998).
- [8] Grodstein, F., Manson, J. E. & Stampfer, M. J. Postmenopausal hormone use and secondary prevention of coronary events in the nurses’ health study: a prospective, observational study. *Annals of Internal Medicine* **135**, 1–8 (2001).
- [9] Grambsch, P. M. & Therneau, T. M. Proportional hazards tests and diagnostics based on weighted residuals. *Biometrika* **81**, 515–526 (1994).
- [10] Epstein, M. P. *et al.* A permutation procedure to correct for confounders in case-control studies, including tests of rare variation. *The American Journal of Human Genetics* **91**, 215–223 (2012).

- [11] Patel, C. J., Bhattacharya, J. & Butte, A. J. An environment-wide association study (EWAS) on type 2 diabetes mellitus. *PLoS One* **5**, e10746 (2010).
- [12] Efron, B. *Large-scale inference: empirical Bayes methods for estimation, testing, and prediction*, vol. 1 (Cambridge University Press, 2012).
- [13] Louis, G. M. B., Smarr, M. M. & Patel, C. J. The exposome research paradigm: an opportunity to understand the environmental basis for human health and disease. *Current Environmental Health Reports* **4**, 89–98 (2017).
- [14] Westfall, P. H. & Young, S. S. *Resampling-based multiple testing: Examples and methods for p-value adjustment*, vol. 279 (John Wiley & Sons, 1993).
- [15] Nyholt, D. R. A simple correction for multiple testing for single-nucleotide polymorphisms in linkage disequilibrium with each other. *The American Journal of Human Genetics* **74**, 765–769 (2004).
- [16] Nicodemus, K. K., Liu, W., Chase, G. A., Tsai, Y.-Y. & Fallin, M. D. Comparison of type I error for multiple test corrections in large single-nucleotide polymorphism studies using principal components versus haplotype blocking algorithms. In *BMC genetics*, vol. 6, S78 (BioMed Central, 2005).
- [17] Tzoulaki, I. *et al.* A nutrient-wide association study on blood pressure. *Circulation* **126**, 2456–2464 (2012).
- [18] Patel, C. J., Cullen, M. R., Ioannidis, J. P. & Butte, A. J. Systematic evaluation of environmental factors: persistent pollutants and nutrients correlated with serum lipid levels. *International Journal of Epidemiology* **41**, 828–843 (2012).
- [19] Patel, C. J. *et al.* Systematic evaluation of environmental and behavioural factors associated with all-cause mortality in the United States National Health and Nutrition Examination Survey. *International Journal of Epidemiology* **42**, 1795–1810 (2013).
- [20] Merritt, M. A. *et al.* Investigation of dietary factors and endometrial cancer risk using a nutrient-wide association study approach in the EPIC and Nurses' Health Study (NHS) and NHS II. *Cancer Epidemiology and Prevention Biomarkers* **24**, 466–471 (2015).
- [21] Willett, W. *Nutritional Epidemiology* (Oxford University Press, 2012).

- [22] Yuan, C. *et al.* Relative validity of nutrient intakes assessed by questionnaire, 24-hour recalls, and diet records as compared with urinary recovery and plasma concentration biomarkers: findings for women. *American Journal of Epidemiology* **187**, 1051–1063 (2017).
- [23] Yuan, C. *et al.* Validity of a dietary questionnaire assessed by comparison with multiple weighed dietary records or 24-hour recalls. *American Journal of Epidemiology* **185**, 570–584 (2017).
- [24] McGinnis, D. P., Brownstein, J. S. & Patel, C. J. Environment-wide association study of blood pressure in the national health and nutrition examination survey (1999–2012). *Scientific Reports* **6**, 30373 (2016).
- [25] Patel, C. J. & Manrai, A. K. Development of exposome correlation globes to map out environment-wide associations. In *Pacific Symposium on Biocomputing Co-Chairs*, 231–242 (World Scientific, 2014).
- [26] Costanza, M. C., Beer-Borst, S., James, R. W., Gaspoz, J.-M. & Morabia, A. Consistency between cross-sectional and longitudinal snp: blood lipid associations. *European Journal of Epidemiology* **27**, 131–138 (2012).
- [27] Xu, Z., Shen, X., Pan, W., Initiative, A. D. N. *et al.* Longitudinal analysis is more powerful than cross-sectional analysis in detecting genetic association with neuroimaging phenotypes. *PLoS One* **9**, e102312 (2014).
- [28] Baghfalaki, T. Bayesian sample size determination for longitudinal studies with continuous response based on different scientific questions of interest. *Journal of Biopharmaceutical Statistics* **29**, 244–270 (2019).
- [29] Yee, J. L. & Niemeier, D. Advantages and disadvantages: Longitudinal vs. repeated cross-section surveys (1996).
- [30] Ericsson, K. A., Hoffman, R. R., Kozbelt, A. & Williams, A. M. *The Cambridge handbook of expertise and expert performance* (Cambridge University Press, 2018).
- [31] Ridker, P. M. *et al.* A randomized trial of low-dose aspirin in the primary prevention of cardiovascular disease in women. *New England Journal of Medicine* **352**, 1293–1304 (2005).
- [32] Manson, J. E. *et al.* A prospective study of aspirin use and primary prevention of cardiovascular disease in women. *JAMA* **266**, 521–527 (1991).

- [33] Iso, H. *et al.* Prospective study of aspirin use and risk of stroke in women. *Stroke* **30**, 1764–1771 (1999).
- [34] Mora, S. & Manson, J. E. Aspirin for primary prevention of atherosclerotic cardiovascular disease: advances in diagnosis and treatment. *JAMA Internal Medicine* **176**, 1195–1204 (2016).
- [35] Lawes, C. M., Bennett, D. A., Lewington, S. & Rodgers, A. Blood pressure and coronary heart disease: a review of the evidence. In *Seminars in Vascular Medicine*, vol. 2, 355–368 (Copyright© 2002 by Thieme Medical Publishers, Inc., 333 Seventh Avenue, New . . . , 2002).
- [36] Huxley, R., Lewington, S. & Clarke, R. Cholesterol, coronary heart disease and stroke: a review of published evidence from observational studies and randomized controlled trials. In *Seminars in Vascular Medicine*, vol. 2, 315–324 (Copyright© 2002 by Thieme Medical Publishers, Inc., 333 Seventh Avenue, New . . . , 2002).
- [37] Hu, Y. *et al.* Mediterranean diet and incidence of rheumatoid arthritis in women. *Arthritis Care and Research* **67**, 597–606 (2015).
- [38] Pun, V. C. *et al.* Prospective study of ambient particulate matter exposure and risk of pulmonary embolism in the nurses’ health study cohort. *Environmental Health Perspectives* **123**, 1265–1270 (2015).
- [39] Wulaningsih, W. *et al.* Investigating nutrition and lifestyle factors as determinants of abdominal obesity: an environment-wide study. *International Journal of Obesity* **41**, 340 (2017).
- [40] Barnard, N. D., Willet, W. C. & Ding, E. L. The misuse of meta-analysis in nutrition research. *JAMA - Journal of the American Medical Association* **318**, 1435–1436 (2017).

Reviewers' comments:

Reviewer #1 (Remarks to the Author):

The authors have answered all questions and comments on the manuscript. No more concerns.

Reviewer #2 (Remarks to the Author):

The revised version of the manuscript contains important edits, additions and clarifications, there are however still major problems in relation to the phenotype definitions and other methods aspects.

The authors have updated the Supplementary Information with a proper cohort characterization. It should be pointed out that the study addresses CHD in women, as the phenotype harbors many sex-specific differences, reg. both symptoms, onset, response to therapy etc. (e.g. <https://www.ncbi.nlm.nih.gov/pubmed/31876924>) I still find it a bit odd that the time window (study period) are the only raw numbers that are presented in the main text on NHS.

Ascertainment of CHD: It is reasonable to validate the diagnosis of MI by inspection of the health records for diagnostic criteria of acute coronary syndrome (ACS), although not updated (ref. 99 vs. e.g. [https://doi.org/10.1016/S0735-1097\(00\)00804-4](https://doi.org/10.1016/S0735-1097(00)00804-4)). It is a bit weak to rely only on the clinical presentation at debut (i.e. ACS) in confirming MI. Moreover, the distinction between nonfatal MI and fatal CHD is neither in line with any conventional classification, nor in line with the structure of the NHS questionnaire. For instance, it is surprising that questions regarding angina pectoris (confirmed by angiogram) and coronary bypass, angioplasty, or stent along myocardial infarction (e.g. section 19. of the 2012 questionnaire) are not included in the quantification of incident cases of CHD. Were patients who started prophylactic (primary and secondary alike) treatment for CHD also evaluated (e.g. section 26 of the 2012 questionnaire)? Also, what diagnoses are fatal CHD a composite of and how many cases were there? Aside from different data sources (health records and autopsies), it is an arbitrary and inconsistent distinction (diagnostically and phenotypically).

Despite revision in Methods, the point with the knowledge graph is not sufficiently clear, especially not since many phenotypes (diamonds) are overlapping. A knowledge graph is likely not the best way to present the result from a literature survey, if that is the purpose? Also, how do the results

from this literature search (120 potential associations to CHD) relate to the 374 exposures that are examined in the study?

Other issues in Methods (comments 6, 25 and 28 cont'd)

The authors include a discussion on reverse causation when justifying that they did not further update dietary records of participants who develop diabetes or cancer during follow-up.

- I do not agree, especially since the authors stress that this is an association study, not unveiling causations. Change in diet can impact risk of CHD development although the motivation to alter dietary patterns are rooted in another way. What would the results be if development of other diseases during follow-up were adjusted for?
- In addition, it is stated that type 2 diabetes is much more dependent on short-term diet, than CHD, which is a radical statement. In the context it is not supported by any evidence. It is certainly not a sound argument for calculating the average of all previous dietary data and not taking time-dose use into account.
- Two annotations have been added to figure 2-a ($i=n$) and ($j=m$), but not a clearer explanation of how to read the figure. E.g. a legend translating the color code, what is the variable "j", in which step was the proportionality assumption assessed and what about the variance inflation factor?
- The response regarding censoring is not clear. What the authors describe is that they censor the individual. But they also state that they do not censor them. Please clarify. Was a similar approach also taken to post-menopausal women, which are known to have a higher risk of CVD?

Comment 14: Regarding modelling age as a spline with four degrees of freedom, please show data and use the most appropriate model selected by e.g. AIC.

Comment 17: I am still not entirely sure if the authors included all the listed covariates in the model, or only a subset on them. "Hence, residual confounding is still possible as a result of ignoring the adjustment for other important risk factors" - Could the authors also comment on which risk factors this could be?

Comment 18: Which p-value is this? And was the PH assumption valid for all covariates of the model?

Comment 34: The authors seem very focused on small effect sizes, yet this is the root cause of a major problem in nutritional research. See e.g. this commentary by Trepanowski and Ioannidis (<https://www.ncbi.nlm.nih.gov/pmc/articles/PMC6054237/>). I therefore disagree with the author's statements that "small effect sizes can be very important". No, they are the effect sizes that are most likely to be influenced by unmeasured confounding. The "explanation" of the hazard ratio is not fulfilling. What is one standard deviation of isorhamnetin? It would be very helpful if the authors could specify the effect size in terms of SI units, and possibly absolute differences between groups, to help the interpretation of the effect.

As I stated earlier this is an ambitious study that implements an environment-wide association study to NHS, and as a method it presents a strategy for providing new evidence in unveiling how diet affects development. But there are still several problems in relation to disease definitions and methodology that are confusing, and possibly not sound.

Reviewer #3 (Remarks to the Author):

I congratulate the authors for their accurate work in accepting our comments and modifying the manuscript.

Paolo Vineis

Reviewer #4 (Remarks to the Author):

This revision has partially addressed the concerns raised previously. Nevertheless, the study remains almost impossible to interpret. Essentially, the authors systematically generate a large number of observations about diet and coronary heart disease (CHD) open to residual confounding and other biases, and then validate these observations in a similar study open to the same biases. The analysis is sophisticated and the figures are attractive, however these positives do not outweigh the fundamental problem that we cannot distinguish whether the observed associations of diet with CHD presented are the result of the dietary items studied or are a reflection of the attributes of the people who eat those dietary items. It is of particular concern that the authors do not validate their observations against higher quality evidence from trials or quasi-experimental studies. The authors also do not consider that their study as well as possibly generating false positives may also generate false negatives.

Some of the original concerns also remain

1. The study is still adjusted for factors that may not be confounders, which could add bias. Nowadays confounders are generally understood to be common causes of exposure and outcome.

Specifically,

Line 19-20: “and control variables such as physical activity, smoking, calorie intake and medication use” However, calorie intake seems like a result of dietary intake, not a cause of dietary intake

Lines 450-4: “We adjusted the analyses for potential risk factors and confounders, including age, body mass index (BMI), physical activity, and total caloric intake s continuous covariates; and ethnicity, smoking status, multivitamin use, vitamin E supplement use, post-menopausal hormone use, aspirin use, high blood pressure, elevated cholesterol, and family history of MI and high blood pressure as categorical variables. We selected the set of confounding variables based on their potential effects on both exposures and the outcome.”

Several of these factors including calorie intake, high blood pressure and elevated cholesterol seem more like a result of dietary intake, not a cause of dietary intake. So the authors should justify their use as confounders in the paper, not just in the response, or repeat the analysis without such adjustment.

2. The abstract still does not give a nuanced view of the achievements of observational epidemiology, uses the word “effect” to refer to “associations” and does not explain whether the study replicated results from randomized controlled trials, quasi experimental studies, or observational studies open to the same biases.

3. There seems to be a misunderstanding about line 51. It says “to overcome the limitations of the traditional observational studies, Environment Wide Association Study (EWAS)...” The main limitations of traditional of observational studies are confounding and selection bias, simply doing more analyses open to confounding and selection bias does not address these key limitations. The authors say they did not mean to imply that EWAS overcomes these biases, so it would be clearer to delete the words “to overcome the limitations of the traditional observational studies”

4. Line 62 says “we consistently recover prior knowledge about diet-disease relationships”. However, the authors seem to be referring to knowledge based on previous observational studies open to the

same biases as this study rather than to the higher quality evidence available from randomized control trials or quasi-experimental studies. To take one example, the paper shows alcohol associated with a lower risk of CHD, however several quasi-experimental Mendelian randomization studies have suggested that alcohol does not protect against CHD

<https://www.ncbi.nlm.nih.gov/pubmed/30955975> and

<https://www.ncbi.nlm.nih.gov/pubmed/25011450> .

Statement on the Revision of ⟨NCOMMS-19-19797⟩

Based on the Referees' Report

Soodabeh Milanlouei Giulia Menichetti Yanping Li Joseph Loscalzo
Walter C Willett Albert-László Barabási

June 3, 2020

Comments by Reviewer #1

The authors have answered all questions and comments on the manuscript. No more concerns.

We would like to thank the Referee for the previous comments - we found the recommendations very helpful, and we are pleased to learn that the Referee found the revised manuscript appropriate for publication.

Comments by Reviewer #2

The revised version of the manuscript contains important edits, additions and clarifications, there are however still major problems in relation to the phenotype definitions and other methods aspects.

We thank the Reviewer for the valuable comments. In the following, we have addressed all of the recommendations offered by the Reviewer. We hope that the revised manuscript will clarify any potential misunderstanding our previous formulation may have created.

1. The authors have updated the Supplementary Information with a proper cohort characterization. It should be pointed out that the study addresses CHD in women, as the phenotype harbors many sex-specific differences, reg. both symptoms, onset, response to therapy etc. (e.g. <https://www.ncbi.nlm.nih.gov/pubmed/31876924>) I still find it a bit odd that the time window (study period) are the only raw numbers that are presented in the main text on NHS.

1. We fully agree with the Referee that there are gender-specific differences regarding the onset and development of CHD. In the manuscript, in lines 94, 330, and 413, we now clarify that this study focuses on CHD among women. Following the Referee's recommendation, in the main text, we now mention in addition to the time window, the number of cases and non-cases (lines 425 and 426). The other cohort characteristics were added to the Supplementary Information (Sections 1 and 2, lines 16 to 35) to improve the readability and respect the journal's word limit.

2. Ascertainment of CHD: It is reasonable to validate the diagnosis of MI by inspection of the health records for diagnostic criteria of acute coronary syndrome (ACS), although not updated (ref. 99 vs. e.g. [https://doi.org/10.1016/S0735-1097\(00\)00804-4](https://doi.org/10.1016/S0735-1097(00)00804-4)). It is a bit weak to rely only on the clinical presentation at debut (i.e. ACS) in confirming MI. Moreover, the distinction between nonfatal MI and fatal CHD is neither in line with any conventional classification, nor in line with the structure of the NHS questionnaire. For instance, it is surprising that questions regarding

angina pectoris (confirmed by angiogram) and coronary bypass, angioplasty, or stent along myocardial infarction (e.g. section 19. of the 2012 questionnaire) are not included in the quantification of incident cases of CHD. Were patients who started prophylactic (primary and secondary alike) treatment for CHD also evaluated (e.g. section 26 of the 2012 questionnaire)? Also, what diagnoses are fatal CHD a composite of and how many cases were there? Aside from different data sources (health records and autopsies), it is an arbitrary and inconsistent distinction (diagnostically and phenotypically).

The definitions of nonfatal MI, fatal CHD, and total CHD (the sum of nonfatal MI and fatal CHD) we used here are not specific to this paper. They have been consistently used in the cohorts of the Nurses' Health Study for over 40 years; the criteria for MI are based on the WHO definition (symptoms, enzyme changes, and EKG changes) and not just clinical presentation at debut. These definitions have been accepted by virtually all major medical journals including the NEJM, JAMA, Lancet, BMJ, Circulation, American Heart Journal and many others (we included multiple references at line 420). The reviewer is correct that nonfatal MI and fatal CHD are in principle representing the same underlying disease process, and for this reason our primary analyses rely on total CHD as the outcome. However, the amount of diagnostic information is often less for fatal CHD (which represent about 1/3 of the total cases) because many people die outside of the hospital or soon after arrival. We, therefore, do not rely simply on CHD on the death certificate to classify as case as fatal CHD; previous history of CHD or compatible symptoms are also required. Because of different diagnostic evidence, we usually secondarily examine nonfatal MI and fatal CHD separately; in virtually all cases, findings have been similar as expected. We do not usually include angina with nonfatal MI or fatal CHD; although the pathology related to atherosclerosis overlaps, acute infarction can also include endothelial dysfunction, and prothrombotic processes.

3. Despite revision in Methods, the point with the knowledge graph is not sufficiently clear, especially not since many phenotypes (diamonds) are overlapping. A knowledge graph is likely not the best way to present the result from a literature survey, if that is the purpose? Also, how do the results from this literature search (120 potential associations to CHD) relate

to the 374 exposures that are examined in the study?

The only purpose of the knowledge graph is to illustrate the extensive previous work on NHS data, and we fully agree with the Referee that it is not the best way to represent the past literature. For this reason, we have a separate *Comparison with the Literature* Section, where we compared our findings not only with the previous studies using NHS data, but also with other existing literature. Not all the 374 exposures examined in this study were evaluated before within NHS data, and consequently, they are absent from the knowledge graph. Thus, there is not a one-to-one relation between the exposures in the knowledge graph and the exposures in our study. A list of associations depicted in Figure 1 in the manuscript is provided in Source Data files (Source Data - Figure 1.xlsx). Finally, prompted by the Referee's comment, we also modified the caption of Figure 1 to more accurately signal the limited role of the knowledge graph.

4. Other issues in Methods (comments 6, 25 and 28 cont'd)

The authors include a discussion on reverse causation when justifying that they did not further update dietary records of participants who develop diabetes or cancer during follow-up.

4.1. I do not agree, especially since the authors stress that this is an association study, not unveiling causations. Change in diet can impact risk of CHD development although the motivation to alter dietary patterns are rooted in another way. What would the results be if development of other diseases during follow-up were adjusted for?

Following the Referee's suggestion, we ran a new analysis in which we adjusted the models for development of intermediate diseases such as Cancer and Diabetes, and continued capturing the dietary records. As summarized by Table R.1, the results are consistent with our original findings in terms of directionality of the relations, but the absolute values of the effect sizes are smaller, and the associations are weaker. In addition to the results provided by this analysis, we would like also to mention that the diagnosis of other disease during follow-up could be a motivation for a person to alter their dietary pattern. If that disease is a risk factor for CHD, such as diabetes, the dietary pattern after disease diagnosis would be confounded by the risk factor (a form of reverse causation by an intermediate risk factor). We agree that one alternative strategy would be to adjust for

Table R.1: EWAS Results While Adjusting for Intermediate Disease Development, Instead of Stopping the Dietary Records Updating.

Type	Exposure	Original Analysis		Adjustment for other diseases' development				
		Hazard ratio	P-value	Hazard ratio	95% CI	P-value of PH	VIF	P-value
Nutrient	Alcohol	0.88	4.98E-11	0.89	(0.86, 0.93)	0.19	1.13	6.38E-09
Nutrient	Added bran from wheat, rice, oat, corn	0.87	5.56E-10	0.91	(0.87, 0.95)	0.09	1.25	3.33E-05
Nutrient	Trans 16:1	1.19	9.59E-10	1.11	(1.05, 1.18)	0.42	2.28	2.99E-04
Nutrient	Discretionary liquid fat	0.86	1.33E-09	0.91	(0.87, 0.96)	0.43	1.54	4.16E-04
Nutrient	Animal MUFA	1.17	1.42E-08	1.14	(1.07, 1.20)	0.69	2.13	7.23E-06
Nutrient	Discretionary solid fat	1.18	3.82E-08	1.12	(1.05, 1.18)	0.29	2.55	3.92E-04
Food	White wine	0.89	4.14E-08	0.89	(0.85, 0.93)	0.85	1.07	9.82E-08
Nutrient	Palmitoleic acid	1.17	4.42E-08	1.13	(1.06, 1.20)	0.62	2.40	7.76E-05
Nutrient	Animal fat	1.17	6.73E-08	1.13	(1.07, 1.20)	0.44	2.30	2.92E-05
Food	Salad/oil and vinegar dressing	0.90	3.41E-07	0.93	(0.90, 0.97)	0.10	1.10	1.15E-03
Food	Yogurt	0.90	1.37E-06	0.93	(0.89, 0.97)	0.02	1.08	1.02E-03
Nutrient	Phytate	0.88	4.01E-06	0.91	(0.86, 0.96)	0.01	2.00	1.07E-03
Nutrient	Stearic acid	1.16	4.25E-06	1.13	(1.06, 1.20)	0.46	2.78	1.21E-04
Nutrient	Carbohydrate from milled wholegrain	0.91	5.03E-06	0.94	(0.90, 0.99)	0.20	1.35	8.62E-03
Nutrient	Sodium	1.14	5.11E-06	1.08	(1.01, 1.14)	0.17	2.40	1.99E-02
Food	Raw carrots	0.91	6.33E-06	0.93	(0.90, 0.97)	0.01	1.11	9.92E-04
Nutrient	Total saturated fat	1.16	1.20E-05	1.11	(1.04, 1.19)	0.38	3.18	1.74E-03
Nutrient	Hydroxyproline	1.12	1.26E-05	1.07	(1.02, 1.13)	0.72	1.69	6.37E-03
Nutrient	Isorhamnetin	0.91	1.59E-05	0.97	(0.93, 1.02)	0.32	1.33	2.46E-01
Food	Liquor	0.92	2.06E-05	0.93	(0.89, 0.96)	0.10	1.06	8.20E-05
Nutrient	Carbohydrate from wholegrain	0.91	2.46E-05	0.94	(0.90, 0.98)	0.09	1.33	7.76E-03
Nutrient	Cereal fiber	0.91	4.04E-05	0.93	(0.89, 0.97)	0.08	1.52	2.25E-03
Food	Red wine	0.91	4.28E-05	0.92	(0.88, 0.97)	0.74	1.04	3.96E-04
Nutrient	Trans 18:2	1.12	4.48E-05	1.02	(0.96, 1.09)	0.81	2.00	4.34E-01
Nutrient	Dietary tocopherols	0.88	5.71E-05	0.91	(0.86, 0.97)	0.26	2.84	5.96E-03
Nutrient	Palmitic acid	1.15	7.01E-05	1.11	(1.04, 1.19)	0.56	3.42	3.03E-03
Nutrient	Dietary folate	0.90	8.33E-05	0.94	(0.89, 0.99)	0.10	1.98	2.54E-02
Food	Doughnuts	1.08	9.84E-05	1.07	(1.03, 1.11)	0.09	1.11	9.07E-04
Nutrient	Beta tocotrienol	0.92	1.07E-04	0.96	(0.91, 1.00)	0.13	1.36	4.48E-02
Nutrient	Plant MUFA	0.90	1.22E-04	0.92	(0.87, 0.97)	0.81	2.06	1.65E-03
Food	Hotdog	1.07	1.26E-04	1.06	(1.02, 1.10)	0.12	1.10	1.95E-03
Food	White bread	1.08	1.61E-04	1.07	(1.03, 1.11)	0.10	1.12	6.53E-04
Nutrient	Natural germ	0.92	1.78E-04	0.96	(0.92, 1.00)	0.06	1.28	7.94E-02
Nutrient	Apigenin	0.92	1.79E-04	0.95	(0.91, 0.99)	0.09	1.16	8.45E-03
Nutrient	Beta tocopherol	0.91	2.73E-04	0.95	(0.90, 1.00)	0.13	1.91	4.16E-02
Nutrient	Natural bran	0.92	2.90E-04	0.95	(0.91, 1.00)	0.26	1.31	3.39E-02
Nutrient	Supplemental selenium	0.92	4.01E-04	0.97	(0.93, 1.02)	0.06	1.53	2.37E-01
Food	Apple juice or cider	1.07	4.49E-04	1.07	(1.03, 1.11)	0.85	1.07	6.06E-04
Nutrient	Dietary manganese	0.91	4.82E-04	0.94	(0.89, 0.99)	0.09	2.05	1.95E-02
Food	Peanuts	0.93	4.92E-04	0.94	(0.90, 0.98)	0.06	1.09	2.32E-03
Nutrient	Alpha tocotrienol	0.92	5.37E-04	0.95	(0.90, 1.00)	0.09	1.61	3.42E-02
Nutrient	Myristic acid	1.10	5.44E-04	1.07	(1.01, 1.13)	0.32	2.01	1.32E-02
Nutrient	Cholesterol	1.10	6.09E-04	1.06	(1.01, 1.12)	0.24	2.03	2.70E-02
Nutrient	Supplemental or fortified folic acid	0.92	6.63E-04	0.97	(0.92, 1.03)	0.05	1.74	3.08E-01
Food	All processed meats	1.07	6.73E-04	1.06	(1.02, 1.10)	0.41	1.15	5.78E-03
Nutrient	Trans 18:1	1.09	6.94E-04	1.05	(1.00, 1.11)	0.92	1.98	6.43E-02
Nutrient	Total manganese	0.92	8.72E-04	0.96	(0.91, 1.01)	0.62	1.64	9.26E-02
Food	Hamburger	1.07	9.53E-04	1.06	(1.02, 1.11)	0.21	1.19	3.81E-03
Food	Beverages with sugar	1.07	9.70E-04	1.07	(1.03, 1.12)	0.93	1.13	5.24E-04
Nutrient	Synthetic vitamin B6	0.94	1.02E-03	0.95	(0.91, 0.99)	0.50	1.09	1.29E-02
Food	Cold breakfast cereal	0.94	1.02E-03	0.95	(0.91, 0.99)	0.18	1.10	8.70E-03
Food	Raisins or grapes	0.93	1.09E-03	0.95	(0.91, 0.99)	0.90	1.12	7.70E-03
Nutrient	Heme iron	1.08	1.17E-03	1.05	(1.00, 1.10)	0.92	1.59	4.80E-02

the diagnosis of these diseases during follow-up. However, because diet is a major determinant of diabetes, controlling for diabetes would be adjusting for variables in the causal pathway between diet and risk of CHD, which would be “over-control”. Thus, stopping the updating of diet at the time of diagnosis of these intermediate diagnoses avoids this over-adjustment. We acknowledge that there is no perfect method for dealing with the issue of confounding and reverse causation by intermediate risk factors, hence we adopted here the commonly used approach [1–4].

4.2. In addition, it is stated that type 2 diabetes is much more dependent on short-term diet, than CHD, which is a radical statement. In the context it is not supported by any evidence. It is certainly not a sound argument for calculating the average of all previous dietary data and not taking time-dose use into account.

We thank the Referee for pointing this out and we made sure to avoid such argument in the main manuscript. Following the Referee’s observation, we investigated the time-dose effect of dietary factors on CHD risk and added a new section to the Supplementary Information (Sections 10), discussing the results. As shown in Figure R.1, the hazard ratios with the cumulative averages are consistent in directionality with hazard ratios calculated considering the time-dose effect, for all those exposures already found statistically significant in the original analysis. We now added pertinent references to the manuscript (line 459) and a new section to the Supplementary Information (Section 4) to justify our choice of cumulative average.

4.3. Two annotations have been added to figure 2-a ($i=n$) and ($j=m$), but not a clearer explanation of how to read the figure. E.g. a legend translating the color code, what is the variable j , in which step was the proportionality assumption assessed and what about the variance inflation factor?

We thank the Referee for pointing this out. We modified the caption of Figure 2-a to provide a clearer explanation of how to read the figure.

4.4. The response regarding censoring is not clear. What the authors describe is that they censor the individual. But they also state that they do not censor

Figure R.1: The estimated Hazard Ratio using cumulative averages versus time-dose effects

them. Please clarify. Was a similar approach also taken to post-menopausal women, which are known to have a higher risk of CVD?

A participant is a non-case until she develops acute MI or fatal CHD as this is the definition of case for this analysis, at which point the follow-up is censored. This is the only censoring that occurs in our analysis. For participants who developed diabetes mellitus or cancer during the follow-up, we did not censor them, but we did not further update their dietary data because these conditions can induce major changes in diet and lifestyle habits. We did adjust for menopausal status and post-menopausal hormone use while continuing follow-up for these women. We did not stop updating diet for women that became menopausal, because there is not good evidence that diet influences the occurrence of menopause, or that the diagnosis of menopause is likely to change women's diets.

5. Comment 14: Regarding modelling age as a spline with four degrees of freedom, please show data and use the most appropriate model selected by e.g. AIC.

Prompted by the Reviewer's request, we added a summary of our results in Section 6 of the Supplementary Material. In Figure SI. 3-a and 3-b, we compared the hazard ratios and P-values from the linear model with the non-linear model where we included a spline of degree four for

age. In both cases, we see that they define a straight line of unit slope, implying that two models are almost identical in terms of the estimated hazard ratios and P-values. In Figure SI. 3-c, we plotted the AIC obtained from each version of the model. The AIC from non-linear models is only slightly smaller than the AIC of linear models. In Figure SI. 3-d, we report the distribution of the percentage of the difference between AIC in the two modeling frameworks. We found that the average of the percentage of the difference in AIC is -0.00225, with a very small variance. Therefore, the change in AIC is not sufficiently large to choose the non-linear model over the linear model, and add to the complexity of the analysis, which results in a significantly higher run-time for each analysis and each iteration of the permutation process.

6. Comment 17: I am still not entirely sure if the authors included all the listed covariates in the model, or only a subset on them. "Hence, residual confounding is still possible as a result of ignoring the adjustment for other important risk factors" - Could the authors also comment on which risk factors this could be?

Of course. Prompted by the Referee, in line 460 to 464, we listed all the covariates for which we adjusted our analysis which includes age, body mass index (BMI), physical activity, and total caloric intake as continuous covariates; and ethnicity, smoking status, multivitamin use, vitamin E supplement use, post-menopausal hormone use, aspirin use, high blood pressure, elevated cholesterol, and family history of MI and high blood pressure. Although we adjusted for many known variables that could potentially confound associations, we cannot be sure that unknown or unmeasured variables could be acting as confounders. Sometimes, the question of confounding by hard-to-measure variables such as socio-economic is raised. This is unlikely to be a major issue because our participants are all health professionals with similar educational background. In re-reading our formulation, we realized that we inadvertently imply that we purposefully left out important confounding factors, which is not the case, hence we have changed the paper to: "Hence, residual confounding by unmeasured variables cannot be excluded" (lines 348 and 349). We thank the Referee for helping us clarify this issue.

7. Comment 18: Which p-value is this? And was the PH assumption valid for all covariates of the model?

We apologize for not being clear on this. In Table 1, the column labeled *P-value of PH* lists the p-values associated with the PH assumption tests. We checked the correlation between the Schoenfeld residuals and survival time, where a correlation close to zero supports the proportional hazards (PH) assumption. For all exposures, except for phytate, the p-value was larger than 0.05. Similar to previous studies [5,6], we are only looking at the violation of PH assumption for the exposure of interest in each test. Because we tested many variables for the PH assumption and only one was significant, this may well be due to chance, but this does not imply that a more detailed examination of the data for phytate may be useful.

8. Comment 34: The authors seem very focused on small effect sizes, yet this is the root cause of a major problem in nutritional research. See e.g. this commentary by Trepanowski and Ioannidis (<https://www.ncbi.nlm.nih.gov/pmc/articles/PMC6054237/>). I therefore disagree with the author's statements that "small effect sizes can be very important". No, they are the effect sizes that are most likely to be influenced by unmeasured confounding. The "explanation" of the hazard ratio is not fulfilling. What is one standard deviation of isorhamnetin? It would be very helpful if the authors could specify the effect size in terms of SI units, and possibly absolute differences between groups, to help the interpretation of the effect.

Given that CHD is the most common cause of death in the US and worldwide, even very small effect sizes can be very important, especially if we can design interventions to modify it. We fully agree that strongest effects are likely to be the most robust. EWAS is, in fact, a screening methodology that ranks the effect sizes and can be used to select those exposures with the most robust signals, discarding others that would (only) pass a single-test analysis. Indeed, Dr. Ioannidis, the co-author of the commentary mentioned by the Referee, is one of the strongest promoters of EWAS in epidemiology, precisely to control for the lack of reproducibility and the small effect sizes (<https://www.ncbi.nlm.nih.gov/pmc/articles/PMC4110965/>).

Regarding the unit of measurements of the effect sizes, given the multiple exposures of different nature, EWAS methodology relies on standard units and not on SI units. All the exposures were standardized to evaluate the effect sizes in comparable scales. Consistently, different exposures have

different standard deviations. The factor e^β is equivalent to the ratio of the hazards between two individuals when they differ for a relative increase equivalent to the geometric standard deviation of the exposure over the population. To help interpret the effect sizes and the possible absolute difference between populations groups, we included Table SI. 7 (Supplementary Information, Section 11), where we calculated the HR of each quintile of exposure intake compared with the first quintile, as a reference group.

As I stated earlier this is an ambitious study that implements an environment-wide association study to NHS, and as a method it presents a strategy for providing new evidence in unveiling how diet affects development. But there are still several problems in relation to disease definitions and methodology that are confusing, and possibly not sound.

We would again like to thank the Referee for evaluating our paper and providing constructive insights that have significantly improved the final manuscript.

Comments by Reviewer #3

I congratulate the authors for their accurate work in accepting our comments and modifying the manuscript.

Paolo Vineis

We are delighted to learn that the Referee found the revised manuscript appropriate for publication

Comments by Reviewer #4

This revision has partially addressed the concerns raised previously. Nevertheless, the study remains almost impossible to interpret. Essentially, the authors systematically generate a large number of observations about diet and coronary heart disease (CHD) open to residual confounding and other biases, and then validate these observations in a similar study open to the same biases.

We thank the Referee for appreciating the improvements of the manuscript. In the following, we address all of constructive comments offered by the Reviewer. We hope that this revised version will clarify any potential misunderstanding our previous formulation may have created.

1. The analysis is sophisticated and the figures are attractive, however these positives do not outweigh the fundamental problem that we cannot distinguish whether the observed associations of diet with CHD presented are the result of the dietary items studied or are a reflection of the attributes of the people who eat those dietary items. It is of particular concern that the authors do not validate their observations against higher quality evidence from trials or quasi-experimental studies. The authors also do not consider that their study as well as possibly generating false positives may also generate false negatives.

We agree with the Referee that validating the results of our analysis using other clinical trials or quasi-experimental studies would be valuable and could further improve confidence in the results. However, to the best of our knowledge, there is no trial or quasi-experimental study comparable to NHS in terms of the measured panel of dietary exposures, or length of follow-up. Other existing studies usually cover a much smaller number of dietary exposures. Given this lack of data, We made great efforts to validate our results with another comparable study (NHS II), pursuing a validation consistent with those commonly applied by the EWAS community [5, 7–11].

Regarding a possible validation with an orthogonal study design, we are not aware of an RCT or MR study that comprehensively studies the many different aspect of diet that we addressed. The closest would be the PREDIMED RCT of Mediterranean diet and risk of CVD conducted in Spain

and the PREDIMED results broadly support our findings. Many hundreds of other randomized trials have examined one variable at a time in relation to CVD risk factors, and their findings are generally consistent with our results.

Some of the original concerns also remain

2. The study is still adjusted for factors that may not be confounders, which could add bias. Nowadays confounders are generally understood to be common causes of exposure and outcome.

Specifically, Line 19-20: "and control variables such as physical activity, smoking, calorie intake and medication use" However, calorie intake seems like a result of dietary intake, not a cause of dietary intake Lines 450-4: "We adjusted the analyses for potential risk factors and confounders, including age, body mass index (BMI), physical activity, and total caloric intake as continuous covariates; and ethnicity, smoking status, multivitamin use, vitamin E supplement use, post-menopausal hormone use, aspirin use, high blood pressure, elevated cholesterol, and family history of MI and high blood pressure as categorical variables. We selected the set of confounding variables based on their potential effects on both exposures and the outcome."

Several of these factors including calorie intake, high blood pressure and elevated cholesterol seem more like a result of dietary intake, not a cause of dietary intake. So the authors should justify their use as confounders in the paper, not just in the response, or repeat the analysis without such adjustment.

The total energy intake can be a confounding variable when studying the relationships between nutrient intake and disease risk. The total energy intake is perhaps largely a consequence of variations in physical activity, body size, and metabolic efficiency, and for these reasons, it might be associated with disease risks [12, 13]. In addition, total caloric intake and nutrient intake are associated because either the nutrients directly contribute to total calories, or because the individuals who have a higher calorie intake also have a higher intake of specific nutrients [14]. Ref. [13] offers an example of total caloric intake as a confounding factor in the relationship between nutrients and coronary artery disease. Therefore, if we are interested in the effect of a specific nutrient, we

must make sure that it is not associated with risk of disease simply because total energy intake is associated with risk of disease. In the literature, it is standard practice to control for total energy intake except when the outcome is weight change.

We adjusted our analysis for high blood pressure and elevated cholesterol because they may, in part, act as confounders since these diagnoses can alter diets, and they are known to be associated with higher risk of CHD [15,16]. In response to the Referee's observation, we added references (lines 463 and 464) to the main manuscript that address the reasons behind our choices. We do acknowledge that high blood pressure and elevated cholesterol can be on a causal pathway between diet and risk of CHD, i.e., be mediators of the effects of diet, and in this case these should not be adjusted for. Thus, they can be both mediators and confounders. There is no perfect solution to this situation, and it may be that we have underestimated the association of some dietary variables that influence blood pressure and blood lipids.

3. The abstract still does not give a nuanced view of the achievements of observational epidemiology, uses the word "effect" to refer to "associations" and does not explain whether the study replicated results from randomized controlled trials, quasi experimental studies, or observational studies open to the same biases.

We agree, hence, following the Referee's recommendation, we now avoid using "effect" in reference to associations in the abstract (lines 16, 18, 20, and 25). Furthermore, following the Referee's recommendation, we now explicitly state that: "Our implementation of EWAS successfully reproduced prior knowledge of diet-CHD associations in the epidemiological literature, and helped us detect new associations that were only marginally studied, opening potential venues for further extensive experimental validation." (Line 20-23).

4. There seems to be a misunderstanding about line 51. It says "to overcome the limitations of the traditional observational studies, Environment Wide Association Study (EWAS)..." The main limitations of traditional of observational studies are confounding and selection bias, simply doing more analyses open to confounding and selection bias does not address these key limitations.

The authors say they did not mean to imply that EWAS overcomes these biases, so it would be clearer to delete the words "to overcome the limitations of the traditional observational studies"

We agree, hence, we followed the Referee's recommendation and modified the manuscript accordingly (line 52).

5. Line 62 says "we consistently recover prior knowledge about diet-disease relationships". However, the authors seem to be referring to knowledge based on previous observational studies open to the same biases as this study rather than to the higher quality evidence available from randomized control trials or quasi-experimental studies. To take one example, the paper shows alcohol associated with a lower risk of CHD, however several quasi-experimental Mendelian randomization studies have suggested that alcohol does not protect against CHD <https://www.ncbi.nlm.nih.gov/pubmed/30955975> and <https://www.ncbi.nlm.nih.gov/pubmed/25011450> .

We agree with the Referee, hence, in the *Comparison with the Literature* section, where we compared our findings with the existing literature, we included both randomized trials and experimental studies. For example, for apigenin, we showed that our finding is consistent with [17], an experimental study, in which the authors concluded that apigenin can regulate cholesterol metabolism *in vivo*, and plays a role in reducing the level of blood fat by promoting cholesterol absorption and conversion, and accelerating reverse cholesterol transport, which can have a direct effect on CHD risk. For α -tocotrienol, we mentioned the work by Prasad [18] focusing on experimental studies regarding the effect of tocotrienols on heart diseases, and we showed that our findings are consistent with this work. Note, however, that evidence from randomized control trials or quasi-experimental studies are not available for all of the nutrients of interest. Regarding alcohol, we now added the references suggested by the Referee, and in response to Referee's observation, we discussed other quasi-experimental Mendelian randomization studies that arrived to a different conclusion on the impact of alcohol consumption (lines 230 to 233). However, many limitations of Mendelian randomization for investigation the alcohol/MI association have been recently described [19].

We would again like to thank the Referee for the constructive insights that have improved the final

manuscript.

References

- [1] Liu, S. *et al.* Whole-grain consumption and risk of coronary heart disease: results from the nurses' health study. *The American journal of clinical nutrition* **70**, 412–419 (1999).
- [2] Hu, F. B. *et al.* Fish and omega-3 fatty acid intake and risk of coronary heart disease in women. *Jama* **287**, 1815–1821 (2002).
- [3] Rimm, E. B. *et al.* Folate and vitamin b6 from diet and supplements in relation to risk of coronary heart disease among women. *Jama* **279**, 359–364 (1998).
- [4] Joshipura, K. J. *et al.* The effect of fruit and vegetable intake on risk for coronary heart disease. *Annals of internal medicine* **134**, 1106–1114 (2001).
- [5] Patel, C. J. *et al.* Systematic evaluation of environmental and behavioural factors associated with all-cause mortality in the united states national health and nutrition examination survey. *International Journal of Epidemiology* **42**, 1795–1810 (2013).
- [6] Eliassen, A. H. *et al.* Use of aspirin, other nonsteroidal anti-inflammatory drugs, and acetaminophen and risk of breast cancer among premenopausal women in the nurses' health study ii. *Archives of internal medicine* **169**, 115–121 (2009).
- [7] Patel, C. J., Cullen, M. R., Ioannidis, J. P. & Butte, A. J. Systematic evaluation of environmental factors: persistent pollutants and nutrients correlated with serum lipid levels. *International Journal of Epidemiology* **41**, 828–843 (2012).
- [8] McGinnis, D. P., Brownstein, J. S. & Patel, C. J. Environment-wide association study of blood pressure in the national health and nutrition examination survey (1999–2012). *Scientific Reports* **6**, 30373 (2016).
- [9] Tzoulaki, I. *et al.* A nutrient-wide association study on blood pressure. *Circulation* **126**, 2456–2464 (2012).
- [10] Wulaningsih, W. *et al.* Investigating nutrition and lifestyle factors as determinants of abdominal obesity: an environment-wide study. *International Journal of Obesity* **41**, 340 (2017).

- [11] Patel, C. J., Bhattacharya, J. & Butte, A. J. An environment-wide association study (EWAS) on type 2 diabetes mellitus. *PLoS One* **5**, e10746 (2010).
- [12] Willett, W. & Stampfer, M. J. Total energy intake: implications for epidemiologic analyses. *American journal of epidemiology* **124**, 17–27 (1986).
- [13] Willett, W. C., Howe, G. R. & Kushi, L. H. Adjustment for total energy intake in epidemiologic studies. *The American journal of clinical nutrition* **65**, 1220S–1228S (1997).
- [14] Arija, V., Abellana, R., Ribot, B. & Ramón, J. M. Biases and adjustments in nutritional assessments from dietary questionnaires. *Nutricion hospitalaria* **31**, 113–118 (2015).
- [15] Lawes, C. M., Bennett, D. A., Lewington, S. & Rodgers, A. Blood pressure and coronary heart disease: a review of the evidence. In *Seminars in Vascular Medicine*, vol. 2, 355–368 (Copyright© 2002 by Thieme Medical Publishers, Inc., 333 Seventh Avenue, New . . . , 2002).
- [16] Huxley, R., Lewington, S. & Clarke, R. Cholesterol, coronary heart disease and stroke: a review of published evidence from observational studies and randomized controlled trials. In *Seminars in Vascular Medicine*, vol. 2, 315–324 (Copyright© 2002 by Thieme Medical Publishers, Inc., 333 Seventh Avenue, New . . . , 2002).
- [17] Zhang, K., Song, W., Li, D. & Jin, X. Apigenin in the regulation of cholesterol metabolism and protection of blood vessels. *Experimental & Therapeutic Medicine* **13**, 1719–1724 (2017).
- [18] Prasad, K. Tocotrienols and cardiovascular health. *Current Pharmaceutical Design* **17**, 2147–2154 (2011).
- [19] Mukamal, K. J., Stampfer, M. J. & Rimm, E. B. Genetic instrumental variable analysis: time to call mendelian randomization what it is. the example of alcohol and cardiovascular disease. *European journal of epidemiology* 1–5 (2019).

REVIEWER COMMENTS

Reviewer #2 (Remarks to the Author):

I thank the authors for having dealt with most of the issues raised; the modifications have clearly added value to paper. There is one exception around the "Ascertainment of CHD", and perhaps the wording in the review was not fully clear. Nice to see the references #95-97 included. However, in the e.g. NEJM reference (#95) non fatal MI is a primary endpoint. In the submitted study non fatal MI is used to define the incidence of CHD. In looking at the NHS questionnaire, many more variables ought to be taken in to account to estimate the true incidence of CHD among NHS participants, e.g. angina pectoris, coronary artery by-pass and medication. So, the major concern here is that specificity of the CHD diagnostic criteria is low and that this will reduce the power of the statistical analysis, incl. reported significant associations to dietary intake. I am not sure the argument around other NHS studies is a good one here.

Statement on the Revision of ⟨NCOMMS-19-19797C⟩

Based on the Referees' Report

Soodabeh Milanlouei Giulia Menichetti Yanping Li Joseph Loscalzo
Walter C Willett Albert-László Barabási

August 2, 2020

Comments by Reviewer #2

I thank the authors for having dealt with most of the issues raised; the modifications have clearly added value to paper. There is one exception around the "Ascertainment of CHD", and perhaps the wording in the review was not fully clear. Nice to see the references 95-97 included. However, in the e.g. NEJM reference (95) non fatal MI is a primary endpoint. In the submitted study non fatal MI is used to define the incidence of CHD. In looking at the NHS questionnaire, many more variables ought to be taken in to account to estimate the true incidence of CHD among NHS participants, e.g. angina pectoris, coronary artery by-pass and medication. So, the major concern here is that specificity of the CHD diagnostic criteria is low and that this will reduce the power of the statistical analysis, incl. reported significant associations to dietary intake. I am not sure the argument around other NHS studies is a good one here.

We are delighted to learn that the Referee found the revisions have added to the value of the paper. We also want to thank for helping to clarify the issue raised earlier. In our ascertainment of CHD, we used the standard World Health Organization criteria for acute myocardial infarction based on elevated specific cardiac enzymes and diagnostic electrocardiogram changes plus hospitalization or fatal coronary heart disease [1,2] (line 445 to 452). Angina Pectoris has an important element of

subjectivity, and by-pass surgery is often elective and depends on how bothersome the symptoms are, the insurance coverage, and variations in medical practice. Furthermore, these outcomes often have an unclear time of onset, making it difficult to be sure that the exposure data (dietary records) precedes the outcome. Our definition of CHD is the accepted standard in this field, and has been used by NHS team in over 200 publications in top medical and cardiology journals, thus it would be inconsistent to alter it. Moreover, the same definition of CHD (non-fatal MI and fatal CHD) has been used in other cohorts, such as the Women’s Health Initiative (WHI) Study [3], the Atherosclerosis Risk in Communities (ARIC) Study [4], and PREDIMED [5]. It is true that a higher number of endpoints might increase power, but in past studies, when we looked at angina and by-pass surgery, we generally saw the same associations although with weaker relative risks, probably due to the lack of specificity [6]. Because power is strongly related to the relative risk, it is possible that changing the criteria would not improve it. The results of the current study show quite good power, as proven by the tight confidence intervals in Table 1, hence we do not need to increase the number of endpoints.

To address the Reviewer’s question, in the Supplementary Information we now acknowledge that less stringent definitions of disease phenotype were used in other cohorts (lines 35 to 39). Prompted by the Referee’s suggestions, we are willing to modify our title from “A Systematic Comprehensive Longitudinal Evaluation of Dietary Factors Associated with Coronary Heart Disease” to “A Systematic Comprehensive Longitudinal Evaluation of Dietary Factors Associated with Major Coronary Heart Disease” to better clarify the scope of our study.

References

- [1] Rose, G. A., Blackburn, H., Gillum, R., Prineas, R. *et al.* *Cardiovascular survey methods.*, vol. 56 (Geneva, Switzerland; WHO, 1982).
- [2] Mendis, S. *et al.* World Health Organization definition of myocardial infarction: 2008–09 revision. *International Journal of Epidemiology* **40**, 139–146 (2011).
- [3] Mossavar-Rahmani, Y. *et al.* Artificially sweetened beverages and stroke, coronary heart disease, and all-cause mortality in the Women’s Health Initiative. *Stroke* **50**, 555–562 (2019).
- [4] Bidulescu, A., Chambless, L. E., Siega-Riz, A. M., Zeisel, S. H. & Heiss, G. Usual choline and betaine dietary intake and incident coronary heart disease: the Atherosclerosis Risk in

Communities (ARIC) study. *BMC cardiovascular disorders* **7**, 20 (2007).

- [5] Garcia-Arellano, A. *et al.* Dietary inflammatory index and incidence of cardiovascular disease in the PREDIMED study. *Nutrients* **7**, 4124–4138 (2015). URL [/pmc/articles/PMC4488776/?report=abstract](https://pubmed.ncbi.nlm.nih.gov/pmc/articles/PMC4488776/?report=abstract)
<https://www.ncbi.nlm.nih.gov/pmc/articles/PMC4488776/>.
- [6] Willett, W. C. *et al.* Relative and Absolute Excess Risks of Coronary Heart Disease among Women Who Smoke Cigarettes. *New England Journal of Medicine* **317**, 1303–1309 (1987). URL <https://pubmed.ncbi.nlm.nih.gov/3683458/>.

REVIEWERS' COMMENTS

Reviewer #2 (Remarks to the Author):

I thank the authors for clarifying the way these data have been used over the years. Sounds like this is based on tradition that possible at one point should change, but not so easy to change for this manuscript I imagine. The new text adds definitely to the clarity of what is going on, although still not indicated what is written on the death certificate of these patients. I am not familiar with US practice, so hard to be constructive. For the title, now we are at the nitty gritty, I think "Major" does not make the purpose of the study clearer, it would reflect the work better just to write "Myocardial Infarction and fatal Coronary Heart disease" in my view. Grand total I think the modifications have indeed improved the paper considerably.

Statement on the Revision of <NCOMMS-19-19797C>

Based on the Referees' Report

Soodabeh Milanlouei Giulia Menichetti Yanping Li Joseph Loscalzo
Walter C Willett Albert-László Barabási

September 26, 2020

Comments by Reviewer #2

I thank the authors for clarifying the way these data have been used over the years. Sounds like this is based on tradition that possible at one point should change, but not so easy to change for this manuscript I imagine. The new text adds definitely to the clarity of what is going on, although still not indicated what is written on the death certificate of these patients. I am not familiar with US practice, so hard to be constructive. For the title, now we are at the nitty gritty, I think "Major" does not make the purpose of the study clearer, it would reflect the work better just to write "Myocardial Infarction and fatal Coronary Heart disease" in my view. Grand total I think the modifications have indeed improved the paper considerably.

We thank the referee for the valuable comments. We changed the title of our manuscript to "A Systematic Comprehensive Longitudinal Evaluation of Dietary Factors Associated with Acute Myocardial Infarction and Fatal Coronary Heart Disease" to further clarify the scope of our work.